# ACTIVE LEARNING BASED STRUCTURAL INFERENCE

## ABSTRACT

In this paper, we propose an active-learning based framework, *Active Learning based Structural Inference* (ALaSI), to infer the existence of directed connections from observed agents' states over a time period in a dynamical system. With the help of deep active learning, ALaSI is competent in learning the representation of connections with relatively small pool of prior knowledge. Moreover, based on information theory, we propose inter- and out-of-scope message learning pipelines, which are remarkably beneficial to the structural inference for large dynamical systems. We evaluate ALaSI on various large datasets including simulated systems and real-world networks, to demonstrate that ALaSI is able to precisely infer the existence of connections in these systems under either supervised learning or unsupervised learning, with better performance than baseline methods.

## 1 INTRODUCTION

Dynamical systems are commonly observed in real-world, including physical systems (Kwapień & Drożdż, 2012; Ha & Jeong, 2021), biological systems (Tsubaki et al., 2019; Pratapa et al., 2020), and multi-agent systems (Brasó & Leal-Taixé, 2020; Li et al., 2022). A dynamical system can be described as a set of three core elements: (a) the state of the system in a time period, including state of the individual agents; (b) the state-space of the system; and (c) the state-transition function (Irwin & Wang, 2017). Knowing these core elements, we can describe and predict how a dynamical system behaves. Yet the three elements are not independent of each other, for example, the evolution of the state is affected by the state-transition function, which suggests that we may predict the future state based on its current state and the entities which affect the agents (i.e. *connectivity*). Moreover, the state-transition function is often deterministic (Katok & Hasselblatt, 1995), which simplifies the derivation of the future state as a Markovian transition function.

However, in most cases, we hardly have access to the connectivity within a given system, or only have limited knowledge about the connectivity. Is it possible to infer the connectivity from observed states of the agents over a time period? We formulate it as the problem of *structural inference*, and several machine learning frameworks have been proposed to address it (Kipf et al., 2018; Webb et al., 2019; Alet et al., 2019; Chen et al., 2021; Löwe et al., 2022; Wang & Pang, 2022). Although these frameworks can accurately infer the connectivity, as they perform representation learning on a fully connected graph, these methods can only work for small systems (up to dozens of agents), and cannot scale well to real-world dynamical systems, for example, with hundreds of agents. Besides that, as we show in the experiment and appendix sections in this work, the integration of prior knowledge about partial connectivity of the system is quite problematic among these methods.

On the other hand, deep active learning (DeepAL) is an emerging branch of research that is used to reduce the cost of annotation while retaining the powerful learning capabilities of deep learning (Ren et al., 2022). This motivates us to explore DeepAL to solve the problem of structural inference. In order to perform structural inference on large dynamical systems, instead of building pools based on batches, we build pools based on agents, and expect the learning framework can consequently infer the existence of directed connections with a little prior knowledge of the connections. Therefore, in this work, based on DeepAL, we propose a novel structural inference framework, namely, **A**ctive **L**earning b**a**sed **S**tructural **I**nference (ALaSI), which is designed for the structural inference of large dynamical systems, and is suitable for the integration of prior knowledge.

ALaSI leverages query strategy with dynamics for agent-wise selection to update the pool with the most informative partial system, which encourages ALaSI to infer the connections efficiently and

accurately with partial prior knowledge on the connectivity (named as 'scope'). Furthermore, based on information theory, ALaSI learns both inter-scope and out-of-scope (OOS) messages from the current scope to distinguish the information which represents connections from agents within the scope and from agents out of the scope, which reserves redundancy when new agents come into scope. Moreover, with oracle such as partial information decomposition (PID) (Williams & Beer, 2010), ALaSI can infer the connectivity even without prior knowledge and be trained in an unsupervised way. We show with extensive experiments that ALaSI can infer the directed connections of dynamical systems with up to 1.5K agents with either supervised learning or unsupervised learning.

## 2 RELATED WORK

**Deep Active learning.** Our framework ALaSI follows the strategy of DeepAL (Ren et al., 2022), which attempts to combine the strong learning capability of deep learning in the context of high-dimensional data processing, as well as the significant potential of active learning (AL) in effectively reducing labeling costs. Within the field of DeepAL, several methods (Gal et al., 2017; Pop & Fulop, 2018; Kirsch et al., 2019; Tran et al., 2019) applied Bayesian deep learning to deal with high-dimensional mini-batch samples with fewer queries in the AL context. To solve the problem of insufficient labeled sample data, Tran et al. (2019) leveraged generative networks for data augmentation, and Wang et al. (2016) assigned pseudo-labels to high-confidence samples to expand the labeled training set. Moreover, Hossain & Roy (2019) and Siméoni et al. (2020) used labeled and unlabeled datasets to combine supervised and semisupervised training with AL methods. There also exist a number of works on how to improve the batch sample query strategy (Shi & Yu, 2019; Kirsch et al., 2019; Zhdanov, 2019; Ash et al., 2020). As we will show, by leveraging the advantages of DeepAL, ALaSI is competent in efficient and accurate inferring the existence of directed connections with a smaller labeled pool of prior knowledge.

**Structural inference.** The aim of structural inference is to accurately reconstruct the connections between the agents in a dynamical system with observational agents' states. Among the wide variety of methods, neural relational inference (NRI) (Kipf et al., 2018) was the first to address the problem of structural inference based on observational agents' states with the help of a VAE operating on a fixed fully connected graph structure. Based on NRI, Webb et al. (2019) proposed factorized neural relational inference (fNRI), extending NRI to multi-interaction systems. Chen et al. (2021) proposed a method with efficient message passing mechanisms (MPM), to increase the accuracy of structural inference for complex systems. Moreover, Alet et al. (2019) proposed a modular meta-learning-based framework that jointly infers the connectivity with higher data efficiency. From the aspect of Granger-causality, amortized causality discovery (ACD) (Löwe et al., 2022) attempted to infer a latent posterior graph from temporal conditional dependence.

In addition to the work mentioned above, several frameworks can also infer the connectivity, but with different problem settings. Various methods (Ivanovic & Pavone, 2019; Graber & Schwing, 2020; Li et al., 2022) were specially designed to infer the connections of dynamic graphs. ARNI (Casadiego et al., 2017) inferred the latent structure based on regression analysis and a careful choice of basis functions. Mutual information was also utilized to determine the existence of causal links and thus could infer the connectivity of dynamical systems (Schreiber, 2000; Wu et al., 2020). Some approaches fitted a dynamics model and then produced a causal graph estimate of the model by using recurrent models (Tank et al., 2021; Khanna & Tan, 2020), or inferred the connections by generating edges sequentially (Johnson, 2017; Li et al., 2018) and others independently pruned the generated edges from an over-complete graph (Selvan et al., 2018).

It is worth mentioning that there exists another branch of research called *graph structure learning*, which aims to jointly learn an optimized graph structure and corresponding graph representations for downstream tasks (Zhu et al., 2021; Fatemi et al., 2021; Jin et al., 2020). Besides that, there is another series of work to reconstruct the structure of directed acyclic graphs (Zheng et al., 2018; Yu et al., 2019; Saeed et al., 2020; Yu et al., 2021). However, because of various reasons, such as the fixed latent space of VAE, or exponential computational efficiency, most of the methods mentioned above are incapable of structural inference on large dynamical systems and have difficulties in the efficient utilization of prior knowledge.

## 3 PROBLEM DEFINITION

In this section, we formally define the problem of *active learning based structural inference* with dynamics. We view a dynamical system $\mathcal{S}$ as $\mathcal{S} = \{\mathcal{V}, \mathcal{E}\}$, in which $\mathcal{V}$ represents the set of $n$ agents in the system: $\mathcal{V} = \{v_i, 1 \leq i \leq n\}$, and $\mathcal{E}$ denotes the directed connections between the agents: $(v_i, v_j) \in \mathcal{E} \subseteq \mathcal{V} \times \mathcal{V}$. We focus on the cases where we have recordings of trajectories of the agents' states: $\mathcal{V} = \{V^t, 0 \leq t \leq T\}$, where $T$ is the total number of time steps, and $V^t$ is the set of features of all the $n$ agents at time step $t$: $V^t = \{v_1^t, v_2^t, \ldots, v_n^t\}$. Based on the trajectories, we aim to infer the existence of directed connections between any agent-pair in the system. The connections are represented as $\mathcal{E} = \{\mathbf{e}_{ij} \in \{0, 1\}\}$, where $\mathbf{e}_{ij} = 1$ (or $= 0$) denotes the existence of connection from agent $i$ to $j$ (or not). We sample a total number of $K$ trajectories of states. With the notations above, we can summarize the dynamics for agents within the system as:

$$v_i^{t+1} = v_i^t + \Delta \cdot \sum_{j \in \mathcal{U}_i} f\Big( \|v_i, v_j\|_\alpha \Big), \tag{1}$$

where $\Delta$ denotes a time interval, $\mathcal{U}_i$ represents the set of agents connected with agent $i$, and $f(\cdot)$ is the state-transition function deriving to dynamics caused by the edge from agent $j$ to $i$, and $\|\cdot, \cdot\|_\alpha$ denotes a distance between the states of two agents.

We state the problem of structural inference as searching for a combinatorial distribution to describe the existence of a directed connection between any agent-pair in the dynamical system. Assume we have two sets of trajectories, the set of trajectories without knowing connectivity $\mathcal{D}_{\text{pool}} = \{\mathcal{V}_{\text{pool}}, \mathcal{E}_\emptyset\}$, and the set of trajectories for training $\mathcal{D}_{\text{train}} = \{\mathcal{V}_{\text{train}}, \mathcal{E}_{\text{train}}\}$, where $\mathcal{E}_\emptyset$ denotes the empty set of connectivity. In this paper, we consider two scenarios: the first scenario is where we have access to the ground-truth of connections $\mathcal{E}$ in the system, and we perform a supervised-learning-based DeepAL with ALaSI, where ALaSI can be defined as:

$$\min_{\mathbf{s}^L:|\mathbf{s}^L|<\mathcal{K}} \mathbb{E}_{e \sim P_{\mathcal{E}_{\text{train}}}, v \sim P_{\mathcal{V}_{\text{train}}}}[\mathcal{L}(e, v; A_{\mathbf{s}^0 \cup \mathbf{s}^L})], \tag{2}$$

where $\mathbf{s}^0$ is the initial pool of $m$ agents chosen from $\mathcal{D}_{\text{train}}$, as well as the connectivity between them, $\mathbf{s}^L$ is the extra pool with budget $\mathcal{K}$, $A$ represents the algorithm of ALaSI, $\mathcal{L}$ denotes the learning objective and we denote $P_x$ as the sampling space of variable $x$. The second scenario is where the ground-truth connectivity is inaccessible during training, and we show that ALaSI is competent to infer the connections in an unsupervised setting with the help of any oracle. In this paper, we choose PID (Williams & Beer, 2010; Lizier et al., 2013) to instantiate the oracle. Thus, instead of having $\mathcal{E}_{\text{train}}$ available in $\mathcal{D}_{\text{train}}$, we leverage PID to calculate the connectivity between the agents in the pool at every round of sampling. In this case, ALaSI can be defined as:

$$\min_{\mathbf{s}^k:|\mathbf{s}^k|<\mathcal{K}} \mathbb{E}_{e \sim P_{\mathcal{E}_{\text{PID}}}, v \sim P_{\mathcal{V}_{\text{train}}}}[\mathcal{L}(e, v; A_{\mathbf{s}^0 \cup \mathbf{s}^k})], \tag{3}$$

where $\mathbf{s}^k = \{\mathcal{V}_{\text{train}}, \mathcal{E}_{\text{PID}}\}$ denotes the pool, in which $\mathcal{E}_{\text{PID}}$ denotes the connections generated by PID operating on the agents in the pool, and the number of agents in $\mathbf{s}^k$ has a budget $\mathcal{K}$. The initial set $\mathbf{s}^0$ is also set up by PID as that of $\mathbf{s}^k$, but with a different size of agents $m$. As the first to perform DeepAL for structural inference, we consider ALaSI with both supervised and unsupervised learning and conduct experiments on both settings, to demonstrate its promising performance.

## 4 METHOD

In this section, we present ALaSI, a scalable structural inference framework based on agent-wise DeepAL. We start by formulating such a learnable framework in Section 4.1. After that, we describe the inter-scope and OOS operations in Section 4.2, which are of great significance to make the framework scalable. Especially, we propose the hybrid loss and the query strategy with dynamics in Sections 4.3 and 4.4, respectively. Last but not least, we discuss the integration of PID into ALaSI in Section 4.5, which enables ALaSI to infer the connectivity with unsupervised learning.

### 4.1 ACTIVE STRUCTURAL INFERENCE WITH DYNAMICS

The basic idea behind ALaSI is to infer the existence of directed connection between two agents with the help of dynamics. According to Equation 1, we may describe it as: the correct inference

of the connectivity enables the algorithm to predict the future states of the agent with smaller error. We formulate the statement as:

$$\underset{\mathcal{U}_i \subseteq \mathcal{V}}{\arg\min} \, \mathbb{E}_{\theta \sim p(\theta | \{\mathcal{V}, \mathcal{E}\})} \mathcal{R}\big(v_i^{t+1}, P(\hat{v}_i^{t+1} | v_i^t, \mathcal{U}_i, \theta)\big), \tag{4}$$

where $\mathcal{U}_i$ represents the agents connected to agent $i$, $\mathcal{R}$ is the loss function to quantify the dynamics prediction error between actual dynamics $v_i^{t+1}$ and predicted dynamics $\hat{v}_i^{t+1}$, and $\theta$ is the parameters of the model. The problem setting in Equation 4 is also widely adopted (Kipf et al., 2018; Webb et al., 2019; Löwe et al., 2022; Wang & Pang, 2022). For small dynamical systems, we can directly follow this formulation and leverage generative models such as a VAE to work on a fully-connected initial graph, in order to infer the connectivity of the whole system. However, for large dynamical systems, it is impractical and unattainable to infer the connectivity in the same way, which is also a common problem observed in the literature on structural inference.

In this work, we extend Equation 4 for large dynamical systems with the help of DeepAL. Unlike previous DeepAL algorithms, which train models on batch-wise selections (Gal et al., 2017; Kirsch et al., 2019; Pop & Fulop, 2018; Tran et al., 2019), we design ALaSI to train on agent-wise selections. The pool consists of features of different agents, and the directed connections between these agents. By training ALaSI on the pool, we try to encourage the framework to capture the statistics to describe the existence of connections between any agent-pair:

$$\underset{\mathcal{U}_i \subseteq \mathcal{D}}{\arg\min} \, \mathbb{E}_{\theta \sim p(\theta | \mathcal{D})} \mathcal{R}\big(v_i^{t+1}, Q(\hat{v}_i^{t+1} | v_i^t, \mathcal{U}_i, \theta)\big). \tag{5}$$

Different from Equation 4, we have a limited scope $\mathcal{D}$ on the available agents and their features, and we can only learn the representation of connections based on current samplings $\mathcal{D}$. However, there possibly simultaneously exit connections between the OOS agents and the agents inside the scope, and discarding the influences of these OOS connections would lead to inaccurate inference results. As a consequence, we need to design the model $Q$ so that it can distinguish the portion of information related to OOS connections and the portion of information coming from connections in the scope, in order to learn the representation of connection precisely and also reserve redundancy for new agents to be added into the pool. We describe the pipeline of ALaSI in Algorithm 1 and Figure 4 in the appendix, and elaborate more details in the following sections.

## 4.2 Inter- / Out-of-Scope Operations

Previous works leveraged a fixed scope on the entire set of agents of the dynamical system, and thus struggled with the curse of scalability (Kipf et al., 2018; Webb et al., 2019; Löwe et al., 2022; Wang & Pang, 2022). To address this issue, we propose a set of inter-/out-of-scope operations in order to make ALaSI scalable. Suppose we have a partial view of $n_p$ agents in the dynamical system ($n_p < n$), and we call the partial view as a scope. For any agent $i$ in the scope, it is possible that it has connections within the scope and also has connections from agents out of the scope simultaneously. We denote $\mathcal{V}_{\text{inter}}^t$ as the set of inter-scope agents' states and $Z_{\text{oos}}^t$ as the summary of out-of-scope (OOS) agents' states for an ego agent $i$ at time-step $t$. $\mathcal{V}_{\text{inter}}^t$ and $Z_{\text{oos}}^t$ share many characteristics: (1) Since both of them represent the features within the same system, the connections between either inter-scope agents or OOS agents and agent $i$ have the same dynamic function as shown in Equation 4; (2) From the perspective of information theory (Kraskov et al., 2004; Belghazi et al., 2018), we may easily reach the statement that: $I(v_i^t; \mathcal{V}_{\text{inter}}^t) \neq 0$ and $I(v_i^t; Z_{\text{oos}}^t) \neq 0$, where $v_i^t$ represents the features of agent $i$ at time step $t$, and $I(\cdot\,;\cdot)$ denotes the mutual information (MI) between two entities. Based on these common characteristics, we reformulate Equation 5 as:

$$\underset{\mathcal{U}_i \subseteq \mathcal{D}}{\arg\min} \, \mathbb{E}_{\theta \sim p(\theta | \mathcal{D})} \mathcal{R}\big(v_i^{t+1}, Q(\hat{v}_i^{t+1} | v_i^t, \mathcal{V}_{\text{inter}}^t, Z_{\text{oos}}^t, \theta)\big). \tag{6}$$

Yet the calculation of $Z_{\text{oos}}^t$ is agnostic, it is necessary to have another set of derivations.

**Proposition 1** *If we assume $Z_{oos}^t$ only captures the information that affects $v_i^t$ and is different from $\mathcal{V}_{inter}^t$, we can have the following statements:*

$$I(\mathcal{V}_{inter}^t; Z_{oos}^t) < I(v_i^{t+1}; Z_{oos}^t), \text{ and } I(\mathcal{V}_{inter}^t; Z_{oos}^t) < I(v_i^{t+1}; \mathcal{V}_{inter}^t). \tag{7}$$

Proposition 1 infers that the MI between $\mathcal{V}_{\text{inter}}^t$ and $Z_{\text{oos}}^t$ is the smallest among the MI between any pair from $\mathcal{V}_{\text{inter}}^t$, $Z_{\text{oos}}^t$ and $v_i^{t+1}$. It also suggests that we may infer information about $Z_{\text{oos}}^t$ from $v_i^{t+1}$.

We prove the proposition in Section A in the appendix. Based on the MI of time series between two sources and its own past state (Lizier et al., 2013), and the Markovian assumption, we have:

$$I(v_i^{t+1}; v_i^t, \mathcal{V}_{\text{inter}}^t, Z_{\text{oos}}^t) = I(v_i^{t+1}; v_i^t) + I(v_i^{t+1}; \mathcal{V}_{\text{inter}}^t | v_i^t) + I(v_i^{t+1}; Z_{\text{oos}}^t | v_i^t, \mathcal{V}_{\text{inter}}^t). \qquad (8)$$

Since MI terms are non-negative by design, the last term on the right of Equation 8 suggests that given $v_i^{t+1}$, we can derive the information about $Z_{\text{oos}}^t$ conditional on $v_i^t$ and $\mathcal{V}_{\text{inter}}^t$.

Therefore, we implement the inter-/out-of-scope representation learner with neural networks and the pipeline of which is shown in the following equations:

$$e_{\text{inter}} = f_{\text{inter2}}([Z_i \odot \mathcal{V}_{\text{inter}}^t, v_i^t]), \text{ where } Z_i = f_{\text{inter1}}([v_i^t, \mathcal{V}_{\text{inter}}^t]), \qquad (9)$$

$$e_{\text{oos}} = f_{\text{oos2}}([f_{\text{oos1}}(v_i^t), e_{\text{inter}}]), \qquad (10)$$

$$e_{\text{out}} = f_{\text{output}}(f_{\text{dynamics}}(e_{\text{inter}}, e_{\text{oos}}), v_i^t), \qquad (11)$$

where $e_{\text{inter}}$ and $e_{\text{oos}}$ are learned inter-/out-of-scope representations ($\mathcal{V}_{\text{inter}}^t$ / $Z_{\text{oos}}^t$), respectively, $[\cdot, \cdot]$ is the concatenation operation, $f_{\text{inter1}}$ is the neural network to learn the existence of connections between agent $i$ and the agents inside the current scope, $Z_i$ represents the connectivity inside the scope with regards to agent $i$, and $\odot$ is the operation to select agents based on connectivity. Suppose we have $\mathcal{K}$ agents in the scope, then $Z_i \in [0, 1]^{\mathcal{K}}$. So for any agent $i$, $j$ in the scope, we have $z_{ij} \in [0, 1]$, representing the connectivity from agent $i$ to agent $j$. In practice, we reparametrize $z_{ij}$ with Gumbel-Softmax (Jang et al., 2017) to enable backpropagation (see Section B.5 in the appendix for implementation). Besides that, $f_{\text{inter2}}$, $f_{\text{oos1}}$, and $f_{\text{oos2}}$ are the neural networks to learn representations of inter-scope messages, OOS embeddings, and OOS messages, respectively. Finally, in Equation 11, we learn the representations for dynamics with $f_{\text{dynamics}}$, and output the future state of agent $i$ ($e_{\text{out}}$) with $f_{\text{output}}$. In addition to the operations mentioned above, we leverage loss functions (in Section 4.3) to encourage ALaSI to extract OOS messages from $v_i^t$ and $\mathcal{V}_{\text{inter}}^t$.

## 4.3 HYBRID LOSS

The loss function of ALaSI has three roles: (a) encouraging the model to learn OOS representations; (b) calculating dynamics error; and (c) estimating the connectivity prediction error.

As mentioned in Section 4.2, we can derive the OOS message $Z_{\text{oos}}^t$ from $v_i^{t+1}$, $v_i^t$ and $\mathcal{V}_{\text{inter}}^t$. Based on the triplet loss (Schultz & Joachims, 2003; Schroff et al., 2015) and Proposition 1, we derive the following loss function to learn the OOS message:

$$\mathcal{L}_{\text{oos}} = \frac{1}{(T-1) \cdot |\mathcal{D}|} \sum_t^{T-1} \sum_{i \in \mathcal{D}} \left[ -I(Z_{\text{oos}}^t; v_i^{t+1}) \right], \qquad (12)$$

where $T$ represents the total count of time-steps, $\mathcal{D}$ represents the current scope, $|\mathcal{D}|$ denotes the number of agents in the scope. (We discuss the derivation in Section A.2 in the appendix.) We implement the calculation and maximization of mutual information with the help of DeepInfo-Max (Hjelm et al., 2019). However, we have to introduce a regularization term to encourage the learned representations of $Z_{\text{oos}}^t$ and $\mathcal{V}_{\text{inter}}^t$ to be independent of each other, and we leverage distance correlation (Székely et al., 2007). As already proved (Székely & Rizzo, 2009; 2012; 2014), the distance correlation between two variables is zero only when two variables are independent of each other. Therefore, we calculate and minimize the distance correlation between $Z_{\text{oos}}^t$ and $\mathcal{V}_{\text{inter}}^t$:

$$\mathcal{L}_{\text{dc}} = \frac{1}{(T-1) \cdot |\mathcal{D}|} \sum_t^{T-1} \sum_{i \in \mathcal{D}} \frac{\text{dCov}^2(Z_{\text{oos}}^t, \mathcal{V}_{\text{inter}}^t)}{\sqrt{\text{dVar}(Z_{\text{oos}}^t)\text{dVar}(\mathcal{V}_{\text{inter}}^t)}}, \qquad (13)$$

where dCov and dVar are the squared sample distance covariance and the distance variance, respectively, and we describe the procedures for calculating these terms in Section B.4.2 in the appendix. Besides that, we also need the loss function for dynamics:

$$\mathcal{L}_{\text{D}} = -\frac{1}{(T-1) \cdot |\mathcal{D}|} \sum_t^{T-1} \sum_{i \in \mathcal{D}} \frac{||v_i^{t+1} - \hat{v}_i^{t+1}||^2}{2\sigma^2} + \text{const}, \qquad (14)$$

where $v^{t+1}$ and $\hat{v}^{t+1}$ are the ground-truth dynamics and predicted dynamics, respectively, and $\sigma$ is the variance. Moreover, we have the loss function for connectivity:

$$\mathcal{L}_{\text{con}} = -\frac{1}{|D|^2} \sum_{i,j \in \mathcal{D}} z_{ij} \log \left( f(\hat{z}_{ij}) \right), \qquad (15)$$

Figure 1: Query strategy with dynamics in ALaSI.

where $f(\cdot)$ denotes the softmax function, $z_i$ and $\hat{z}_i$ represent the ground-truth connectivity and predicted connectivity in the scope, respectively. With the proposed terms above, we can summarize the hybrid loss function $\mathcal{R}$ as:

$$\mathcal{R} = \alpha \cdot \mathcal{L}_D + \beta \cdot \mathcal{L}_{con} + \gamma \cdot \mathcal{L}_{oos} + \eta \cdot \mathcal{L}_{dc}, \tag{16}$$

where $\alpha$, $\beta$, $\gamma$ and $\eta$ are the weights for the loss terms, trying to match the scales of the last three loss terms with the dynamic loss $\mathcal{L}_D$. We state the details of loss terms in Section B.4 in the appendix.

### 4.4 QUERY WITH DYNAMICS

Interestingly, active learning is also called "query learning" in the statistics literature (Settles, 2009), which indicates the importance of query strategies in the algorithms of active learning. Query strategies are leveraged to decide which instances are most informative and aim to maximize different performance metrics (Settles, 2009; Konyushkova et al., 2017). Query strategies select queries from the pool and update the training set accordingly.

In this work, we propose a novel pool-based strategy: *Query with Dynamics*, which selects queries of $\mathcal{K}$ agents with the largest dynamics prediction error $\mathcal{L}_D$ from the pool $\mathcal{D}_{pool}$, and then we update training set $\mathcal{D}$ with the features and connectivity of $\mathcal{K}$ agents from $\mathcal{D}_{train}$. If we have no access to the connectivity as in unsupervised learning, we run PID to align directed connections to the agents in pool $\mathcal{D}$ with additional $\mathcal{K}$ agents (as shown in lines 27-30 in Algorithm 1). We describe the query strategy in Figure 1 and Algorithm 2 in the appendix. It is notable that despite we have labels on the existence of connections, we do not query agents on it. The reason is that we firstly follow the characteristic of dynamical systems (Equation 4), where the wrong alignment of connections leads to large dynamics error $\mathcal{L}_D$. We secondly try to reserve redundancy for unsupervised learning cases, where ALaSI has no access to ground-truth connections. In this case, we ought to use alternative algorithms as an oracle, such as PID, to estimate the existence of connections and build $\mathcal{D}_{train}$. However, it may be risky that the oracle has a strong bias on the set for training $\mathcal{D}$, and thus errors in this set are unavoidable. As a result, we query agents from the entire pool $\mathcal{D}_{pool}$ according to their dynamics error $\mathcal{L}_D$, thus wrong connections would be recognized by our query strategy.

### 4.5 STRUCTURAL INFERENCE WITH PID

As mentioned above, it is possible that we have no access to the ground-truth connectivity of the dynamical system. ALaSI manages to infer the connections with the help of an oracle: Partial Information Decomposition (PID) (Williams & Beer, 2010; Lizier et al., 2013). The PID framework decomposes the information that a source of variables provides about a destination variable (Lizier et al., 2013). In our cases to infer the existence of directed connections between a pair of agents $i$ and $j$, we follow (Pratapa et al., 2020) with temporal ordering to infer the direction of connections, and we summarize them in Algorithm 3 in the appendix.

With the help of PID, ALaSI can now infer the existence of directed connections even without any prior knowledge about the connectivity, which broadens the application scenarios of ALaSI. We argue that it is possible to use other methods as an oracle for ALaSI, such as pure mutual-information-based methods, SCODE (Matsumoto et al., 2017) or even classic VAE-based structural inference methods (Kipf et al., 2018; Webb et al., 2019; Alet et al., 2019; Löwe et al., 2022), which shows a high ability of adaption and wide application scenario of ALaSI.

## 5 EXPERIMENTS

We test ALaSI on seven different dynamical systems, including simulated networks and real-world gene regulatory networks (GRNs). We also present the ablation study on ALaSI, to show the performance gain with OOS operation and the importance of query of dynamics. Implementation details

can be found in Section B in the appendix. Besides that, we include additional experiments on the integration of prior knowledge with unsupervised learning in Section C in the appendix.

**Datasets.** We firstly test our framework on physical simulations of spring systems, which is also mentioned in (Kipf et al., 2018). Different from that in (Kipf et al., 2018), we sample the trajectories of balls in the system with fixed connectivity, but with different initial conditions. We sample the trajectories by varying the number of balls: {50, 100, 200, 500}, and we name the corresponding datasets as: "Springs50", "Springs100", "Springs200", and "Springs500". Moreover, we collect three real-world GRNs from literature, namely single cell dataset of embryonic stem cells (ESC) (Biase et al., 2014), a cutoff of Escherichia coli microarray data (E. coli) (Jozefczuk et al., 2010), and a cutoff of Staphylococcus aureus microarray data (S. aureus) (Marbach et al., 2012). And the three GRNs have 96, 1505 and 1084 agents, respectively.

**Baselines and metrics.** We compare ALaSI with the state-of-the-art models:

- NRI (Kipf et al., 2018): a variational-auto-encoder model for relational inference.
- fNRI (Webb et al., 2019): an NRI-based model which factorizes the inferred latent interaction graph into a multiplex graph, allowing each layer to encode for a different connection-type.
- MPM (Chen et al., 2021): an NRI-based method with a relation interaction mechanism and a spatio-temporal message passing mechanism.
- ACD (Löwe et al., 2022): a variational model that leverages shared dynamics to infer causal relations across samples with different underlying causal graphs.
- MPIR (Wu et al., 2020): a model based on minimum predictive information regularization.
- PID (Williams & Beer, 2010): computes the ratio between unique mutual information between any agent-pair in the system, and aligns connections according to the ranking.

Despite NRI, fNRI, MPM and ACD being originally designed to infer the connectivity with unsupervised learning, we follow the description in their paper and only train the encoders to show their results of supervised learning. We describe the implementation details of the baseline methods in Section B.6. We demonstrate our evaluation results with the area under the receiver operating characteristic (AUROC), which demonstrates the model's ability to discriminate between cases (positive examples) and non-cases (negative examples), and in this paper, it is used to make clear the method's ability to distinguish actual connections and non-connections.

## 5.1 EXPERIMENTAL RESULTS OF SUPERVISED LEARNING

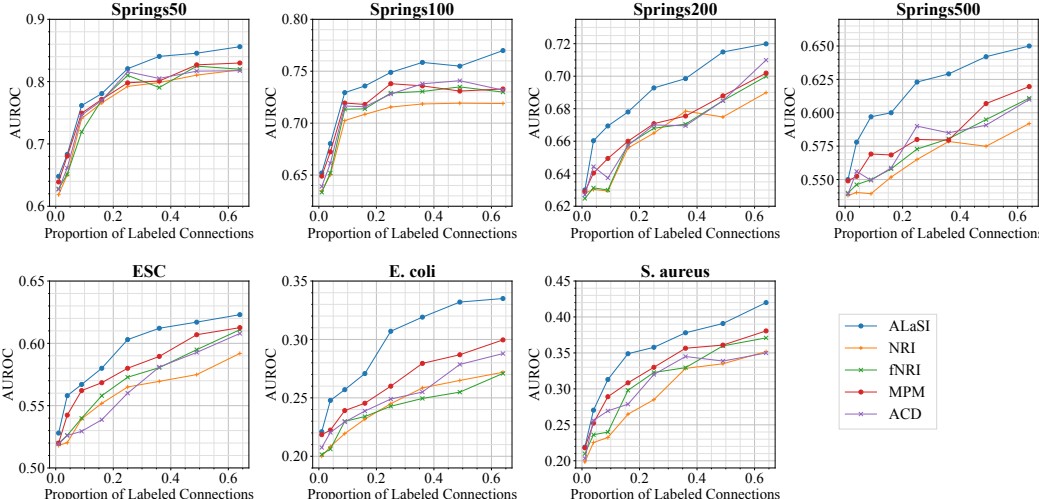

Figure 2: Averaged AUROC results of ALaSI and baseline methods as a function of the proportion of labeled connections. Baseline methods are modified to be trained in a supervised way.

We firstly train ALaSI and baseline methods with supervised learning. It is worth mentioning that despite our efforts, we did not find an approach to train MPIR and PID in a supervised way without

violating their inference mechanisms. For the rest of the baseline methods, we follow the description in their paper and only train the encoders on the partial knowledge of connections. The experimental results of ALaSI and baseline methods are shown in Figure 2. We report the results as the averaged AUROC values of ten runs and as a function of the proportion of labeled connections. And the number of labeled connections is calculated as the square of the number of agents in the scope, where we mark both connections and non-connections as labeled. We subtract the number of labeled connections with the square of the total number of agents in the system to obtain the proportion of labeled connections. Each sub-figure corresponds to the experimental results on a specific dataset.

As shown in Figure 2, the results of baseline methods are positively affected by the proportion of labeled connections during training, and only MPM is marginally better than the other baseline methods on most of the datasets. The rest of the baselines perform almost equally. The averaged AUROC values of ALaSI are also positively correlated with the proportion of labeled connections, but the results are much better than any of the baselines. Although ALaSI is only marginally better than any of the baselines on the datasets of "Springs50" and "Springs100" when the proportion of labeled connections is relatively small (smaller than 0.1), ALaSI outperforms baselines greatly when the proportion of labeled connections is greater than 0.2 on these datasets. ALaSI also infers connectivity with remarkably higher accuracy than baseline methods on the rest of the datasets.

Moreover, we also observe that ALaSI learns the connectivity of the dynamical systems more efficiently than baselines. For example, as shown in the experimental results on all of the datasets except "Springs200", with only 60% of the prior knowledge, ALaSI reaches higher inference accuracy than any baseline methods operating with 80% of the prior knowledge. And this phenomenon is more remarkable in "Springs100", "Springs500" and "E. coli", where ALaSI outperforms baselines with only 50% of the prior knowledge. Thanks to DeepAL and query with dynamics, ALaSI can update the labeling pool with the most informative addition of agents. Besides that, the inter-scope and OOS operations encourage the model to learn connections within the scope and meanwhile also reserve redundancy for possible OOS connections. Consequently, ALaSI is able to learn the connectivity of dynamical systems with less prior knowledge under supervised learning.

## 5.2 Experimental Results of Unsupervised Learning

We report the final averaged AUROC values of ALaSI and baseline methods under unsupervised learning from ten runs in Table 1, and the averaged training time in Table 2. We can observe from Table 1 that all of the methods unsurprisingly perform worse than themselves in supervised learning, which is also stated in (Kipf et al., 2018; Chen et al., 2021). ALaSI performs better than any of the baseline methods on all of the datasets with large margins (up to 0.171), which certainly verifies the inference accuracy of ALaSI on the unsupervised structural inference of large dynamical systems.

Table 1: Averaged AUROC results of baseline methods and ALaSI with unsupervised learning.

| Method | Springs50 | Springs100 | Springs200 | Springs500 | ESC | E. coli | S. aureus |
|--------|-----------|------------|------------|------------|-------|---------|-----------|
| NRI | 0.617 | 0.552 | 0.537 | 0.511 | 0.392 | 0.156 | 0.351 |
| fNRI | 0.621 | 0.567 | 0.540 | 0.516 | 0.398 | 0.154 | 0.351 |
| MPM | 0.631 | 0.590 | 0.544 | 0.518 | 0.402 | 0.170 | 0.375 |
| ACD | 0.620 | 0.589 | 0.539 | 0.515 | 0.387 | 0.162 | 0.369 |
| MPIR | 0.497 | 0.444 | 0.420 | 0.411 | 0.306 | 0.151 | 0.331 |
| PID | 0.678 | 0.630 | 0.592 | 0.547 | 0.451 | 0.195 | 0.378 |
| ALaSI | **0.735** | **0.698** | **0.661** | **0.632** | **0.573** | **0.234** | **0.392** |

Moreover, averaged training time of ALaSI and baseline methods is shown in Table 2. It is worth mentioning that most of the baseline methods are trained on multiple GPU cards when the dataset has more than 100 agents, while ALaSI is trained on a single GPU card. Experimental settings with details may refer to Section B.1. The averaged training time of ALaSI is only longer than MPIR across all of the datasets, while much more accurate than MPIR.Although the AUROC values of PID are the highest among baseline methods, its operation time is much longer than the rest, and it is nevertheless less accurate than ALaSI. Compared with the rest of the baselines, thanks to the query strategy with dynamics and the OOS operation, ALaSI manages to infer the connections for large dynamical systems with higher efficiency even with unsupervised learning.

Table 2: Averaged training time (in H) of baseline methods and ALaSI with unsupervised learning.

| Method | Springs50 | Springs100 | Springs200 | Springs500 | ESC | E. coli | S. aureus |
|--------|-----------|------------|------------|------------|------|---------|-----------|
| NRI    | 29.2      | 40.6       | 57.1       | 85.1       | 39.4 | 118.6   | 101.7     |
| fNRI   | 31.0      | 49.0       | 58.0       | 86.8       | 42.0 | 121.4   | 105.3     |
| MPM    | 35.9      | 51.6       | 57.4       | 85.6       | 44.1 | 124.0   | 105.9     |
| ACD    | 49.0      | 82.4       | 63.9       | 90.0       | 80.4 | 130.9   | 113.5     |
| MPIR   | **12.6**  | **20.7**   | **42.0**   | **51.5**   | **19.5** | **65.1** | **47.6** |
| PID    | 51.6      | 100.2      | 151.0      | 183.4      | 89.3 | 267.1   | 230.8     |
| ALaSI  | 25.5      | 33.8       | 46.1       | 60.3       | 37.2 | 87.0    | 72.9      |

## 5.3 ABLATION STUDY

We conduct ablation studies on the effectiveness of query with dynamics, as well as OOS operation. We modify ALaSI into: (a) ALaSI-ran: where we replace the query with dynamics strategy with a random sampling strategy on agents; and (b) ALaSI-no OOS: where we remove the pipeline for OOS representation learning and the corresponding terms in the loss function. We report the results of unsupervised learning, which we believe is closer to real-world scenarios, and report the averaged AUROC results of these variants as a function of the proportion of labeled connections by PID.

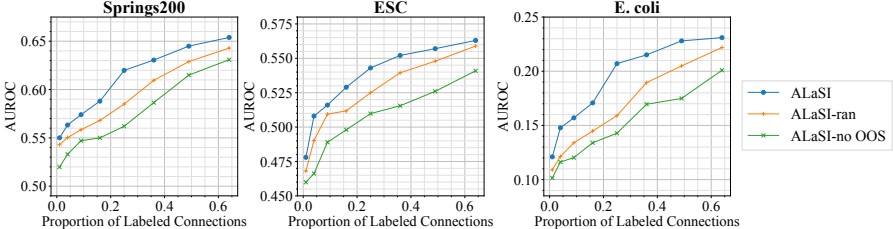

Figure 3: Averaged AUROC results of ALaSI, ALaSI-ran and ALaSI-no OOS as a function of the proportion of labeled connections under unsupervised learning.

As shown in Figure 3, ALaSI with query strategy with dynamics and OOS operation outperforms its variants, ALaSI-random and ALaSI-no OOS. Despite the inference accuracy of all these methods increasing when a large portion of agents are labeled, we observe that ALaSI converges much faster than the other two methods. Besides that, OOS operation is of great importance to the design of a scalable structural inference method. It is commonly observed among the subplots that ALaSI-no OOS can only learn about the representations of connections within the scope and cannot extrapolate onto OOS connections, which results in an almost linear dependence between AUROC and the proportion of labeled connections. Therefore, the query strategy with dynamics and the OOS operation of ALaSI effectively encourage faster convergence under unsupervised settings.

## 6 CONCLUSION

This paper has introduced ALaSI, a scalable structural inference framework based on DeepAL. The query with dynamics encourages the framework to select the most informative agents to be labeled based on dynamics error, and thus leads to faster convergence. The OOS operation enables the framework to distinguish inter-scope messages and OOS messages based on the current view of the partial system, which on the other hand promotes the scalability of ALaSI. The experimental results on the seven datasets have validated the scalability and inference accuracy of ALaSI. The experiments under supervised settings suggest the possibility of leveraging ALaSI to infer the connectivity of dynamical systems with less prior knowledge. Moreover, the experiments under unsupervised settings demonstrate the broad application scenarios of ALaSI to infer the connectivity even without prior knowledge. Future research includes the structural inference based on causality and structural inference for systems with changing agents and connections.

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

## A PROOFS

### A.1 PROOF OF PROPOSITION 1

We prove Proposition 1 in this section. Since we assume the independence between $\mathcal{V}_{\text{inter}}^t$ and $Z_{\text{oos}}^t$, based on the definition of mutual information between two independent variables, we can easily get to the first statement:

$$I(\mathcal{V}_{\text{inter}}^t; Z_{\text{oos}}^t) \approx 0. \tag{17}$$

Moreover, from the proposed PI-diagram of information in a target decomposed from three source variables (Lizier et al., 2013), we have the following statement:

$$I(v_i^{t+1}; v_i^t, \mathcal{V}_{\text{inter}}^t, Z_{\text{oos}}^t) > 0. \tag{18}$$

We refer to Figure 3 in (Lizier et al., 2013) and search for the terms related to $X_{\text{inter}}^t$ and $Z_{\text{oos}}^t$:

$$I(v_i^{t+1}; \mathcal{V}_{\text{inter}}^t) = \{\mathcal{V}_{\text{inter}}^t\} + \{\mathcal{V}_{\text{inter}}^t\}\{v_i^t, Z_{\text{oos}}^t\} + \{v_i^t\}\{\mathcal{V}_{\text{inter}}^t\}\{Z_{\text{oos}}^t\} + \{\mathcal{V}_{\text{inter}}^t\}\{Z_{\text{oos}}^t\} + \\ \{v_i^t\}\{\mathcal{V}_{\text{inter}}^t\} \gg 0, \tag{19}$$

$$I(v_i^{t+1}; Z_{\text{oos}}^t) = \{Z_{\text{oos}}^t\} + \{Z_{\text{oos}}^t\}\{v_i^t, \mathcal{V}_{\text{inter}}^t\} + \{v_i^t\}\{\mathcal{V}_{\text{inter}}^t\}\{Z_{\text{oos}}^t\} + \{\mathcal{V}_{\text{inter}}^t\}\{Z_{\text{oos}}^t\} + \\ \{v_i^t\}\{Z_{\text{oos}}^t\} \gg 0, \tag{20}$$

where $\{\cdot\}\{\cdot\}$ denotes the redundant information in the two sources, $\{\cdot\}\{\cdot\}\{\cdot\}$ denotes the redundant information in the three sources, $\{\cdot\}$ represents the unique information in the single source, and $\{\cdot, \cdot\}$ is the synergistic information from the sources. We summarize the results from Equation 17 to 20, and can derive to:

$$I(\mathcal{V}_{\text{inter}}^t; Z_{\text{oos}}^t) < I(v_i^{t+1}; Z_{\text{oos}}^t), \text{ and } I(\mathcal{V}_{\text{inter}}^t; Z_{\text{oos}}^t) < I(v_i^{t+1}; \mathcal{V}_{\text{inter}}^t), \tag{21}$$

which is Proposition 1.

### A.2 DERIVATION OF OOS LOSS FUNCTION

We describe the derivation procedure for Equation 12 in this section. As mentioned in Section 4.2, we can derive the OOS message $Z_{\text{oos}}^t$ from $v_i^{t+1}$, $v_i^t$ and $\mathcal{V}_{\text{inter}}^t$. Based on the triplet loss (Schroff et al., 2015; Schultz & Joachims, 2003) and Proposition 1, we derive the following loss function to learn OOS message:

$$\mathcal{L}_{\text{oos}} = \frac{1}{(T-1) \cdot |\mathcal{D}|} \sum_t^{T-1} \sum_{i \in \mathcal{D}} \big[ I(\mathcal{V}_{\text{inter}}^t; Z_{\text{oos}}^t) - I(v_i^{t+1}; Z_{\text{oos}}^t) + \alpha_1 + \\ I(\mathcal{V}_{\text{inter}}^t; Z_{\text{oos}}^t) - I(v_i^{t+1}; \mathcal{V}_{\text{inter}}^t) + \alpha_2 \big], \tag{22}$$

where $T$ represents the total count of time-steps, $\mathcal{D}$ represents the current scope, $|\mathcal{D}|$ denotes the number of agents in the scope, and $\alpha_1$ and $\alpha_2$ are margins to regulate the distance between two pairs of mutual information, respectively, in order to encourage larger values of $I(v_i^{t+1}; Z_{\text{oos}}^t)$ and $I(v_i^{t+1}; \mathcal{V}_{\text{inter}}^t)$ compared to $I(\mathcal{V}_{\text{inter}}^t; Z_{\text{oos}}^t)$. It is notable that $Z_{\text{oos}}^t$ and $\mathcal{V}_{\text{inter}}^t$ are calculated according to every agent in the scope, respectively. We omit the subscript of $Z_{\text{oos}}^t$ and $\mathcal{V}_{\text{inter}}^t$ for agent $i$ in Equation 22 for concise. Then we can derive:

$$\mathcal{L}_{\text{oos}} = \frac{1}{(T-1) \cdot |\mathcal{D}|} \sum_t^{T-1} \sum_{i \in \mathcal{D}} \big[ I(\mathcal{V}_{\text{inter}}^t; Z_{\text{oos}}^t) - I(v_i^{t+1}; Z_{\text{oos}}^t) + \alpha_1 + I(\mathcal{V}_{\text{inter}}^t; Z_{\text{oos}}^t) - \\ I(v_i^{t+1}; \mathcal{V}_{\text{inter}}^t) + \alpha_2 \big]$$

$$= \frac{1}{(T-1) \cdot |\mathcal{D}|} \sum_t^{T-1} \sum_{i \in \mathcal{D}} \big[ H(Z_{\text{oos}}^t) - H(Z_{\text{oos}}^t | \mathcal{V}_{\text{inter}}^t) - \big( H(Z_{\text{oos}}^t) - H(Z_{\text{oos}}^t | v_i^{t+1}) \big) + \alpha_1 + \\ H(\mathcal{V}_{\text{inter}}^t) - H(\mathcal{V}_{\text{inter}}^t | Z_{\text{oos}}^t) - \big( H(\mathcal{V}_{\text{inter}}^t) - H(\mathcal{V}_{\text{inter}}^t | v_i^{t+1}) \big) + \alpha_2 \big]$$

$$= \frac{1}{(T-1) \cdot |\mathcal{D}|} \sum_t^{T-1} \sum_{i \in \mathcal{D}} \big[ H(Z_{\text{oos}}^t | v_i^{t+1}) - H(Z_{\text{oos}}^t | \mathcal{V}_{\text{inter}}^t) + \alpha_1 + H(\mathcal{V}_{\text{inter}}^t | v_i^{t+1}) - \\ H(\mathcal{V}_{\text{inter}}^t | Z_{\text{oos}}^t) + \alpha_2 \big].$$

We assume $Z_{\text{oos}}^t$ and $\mathcal{V}_{\text{inter}}^t$ are independent of each other, and we can reformulate the equation as:

$$
\begin{aligned}
\mathcal{L}_{\text{oos}} =& \frac{1}{(T-1) \cdot |\mathcal{D}|} \sum_t^{T-1} \sum_{i \in \mathcal{D}} \big[ H(Z_{\text{oos}}^t | v_i^{t+1}) - H(Z_{\text{oos}}^t) + \alpha_1 + H(\mathcal{V}_{\text{inter}}^t | v_i^{t+1}) - \\
& H(\mathcal{V}_{\text{inter}}^t) + \alpha_2 \big] \\
=& \frac{1}{(T-1) \cdot |\mathcal{D}|} \sum_t^{T-1} \sum_{i \in \mathcal{D}} \big[ - I(Z_{\text{oos}}^t; v_i^{t+1}) + \alpha_1 + - I(\mathcal{V}_{\text{inter}}^t; v_i^{t+1}) + \alpha_2 \big].
\end{aligned}
$$

Since the mutual information between two fixed variables are certain, we omit the second term in the above derivation. Besides that, since the target is the minimization, the constant term has no effect on the formulation. As a result, we can obtain:

$$
\mathcal{L}_{\text{oos}} = \frac{1}{(T-1) \cdot |\mathcal{D}|} \sum_t^{T-1} \sum_{i \in \mathcal{D}} \big[ - I(Z_{\text{oos}}^t; v_i^{t+1}) \big],
$$

which is the formulation in Equation 12. As a result, we only need to minimize $-I(Z_{\text{oos}}; v_i^{t+1})$, and we can implement it with DeepInfoMax (Hjelm et al., 2019) algorithm. DeepInfoMax maximizes the mutual information between input data and learned high-level representations with the help of global and local information.

## B    IMPLEMENTATION

### B.1    GENERAL SETTINGS

We implement ALaSI in PyTorch Paszke et al. (2019) with the help of Scikit-Learn Pedregosa et al. (2011) to calculate various metrics. We run experiments of ALaSI on a single NVIDIA Tesla V100 SXM2 graphic card, which has 32 GB graphic memory and 5120 NVIDIA CUDA Cores. We attach our pseudocode and implementation as the supplementary document to this paper. During training, we set batch size as 64 for datasets which have less than 100 agents, for those equal or more than 100 agents, we set batch size as 16. We train our ALaSI model with 500 epochs for each updated label pool on every dataset.

As for baseline methods, since the training under supervised settings only requires the encoder of the model, which demands moderate space, we managed to run the methods on a single NVIDIA Tesla V100 SXM2 graphic card, and the batch sizes are the same as ALaSI. However, when it came to unsupervised learning, the computational requirement of variational auto-encoder-based methods increased significantly. As a result, in order to run these methods on scalable datasets with more than 100 agents, we use "DistributedDataParallel" of PyTorch to enable the parallel training of these models. And we ran these methods on four NVIDIA Tesla V100 SXM2 graphic cards, with a batch size of 128. For the experiments on datasets with less than 100 agents, we just ran the baselines on a single NVIDIA Tesla V100 SXM2 graphic card with a batch size of 64. For MPIR, since the model is super small and the computational requirement is the smallest among all of the baselines, we ran it on a single NVIDIA Tesla V100 SXM2 graphic card with a batch size of 64. For all of the experiments, we train ALaSI with a learning rate of 0.0005.

### B.2    HYPER-PARAMETERS

We have the following hyper-parameters: initial sample size $m$, query size $\mathcal{K}$, number of epochs $E$, number of selection rounds $N$, variance $\sigma$ of $\mathcal{L}_{\text{dc}}$, weights $\alpha$, $\beta$, $\gamma$, $\xi$ in hybrid loss, and proportion of rank in PID $\eta$. We utilized grid search for the rough values of these hyper-parameters, and show them in Table 3.

### B.3    DETAILS OF PIPELINES

In this section, we firstly demonstrate the general pipeline of ALaSI in Algorithm 1 and Figure 4. Then we show the implementation in Algorithm 4, which is followed by the description of PID algorithm in ALaSI in Algorithm 3.

Table 3: Hyper parameter choices for every dataset.

| DATASET | $m$ | $\mathcal{K}$ | $E$ | $N$ | $\sigma$ | $\alpha$ | $\beta$ | $\gamma$ | $\xi$ | $\eta$ |
|---|---|---|---|---|---|---|---|---|---|---|
| Springs50 | 5 | 0.10 | 500 | 12 | 0.0008 | 0.05 | 0.8 | 20 | 2 | 0.3 |
| Springs100 | 5 | 0.05 | 500 | 15 | 0.0008 | 0.02 | 0.8 | 30 | 2 | 0.3 |
| Springs200 | 10 | 0.04 | 500 | 20 | 0.0008 | 0.02 | 0.5 | 20 | 3 | 0.2 |
| Springs500 | 20 | 0.02 | 600 | 30 | 0.0008 | 0.02 | 0.6 | 40 | 3 | 0.2 |
| ESC | 5 | 0.05 | 500 | 20 | 0.0008 | 0.02 | 0.5 | 50 | 2 | 0.2 |
| E. coli | 20 | 0.02 | 600 | 50 | 0.0008 | 0.01 | 0.4 | 40 | 3 | 0.3 |
| S. aureus | 20 | 0.02 | 600 | 50 | 0.0008 | 0.01 | 0.4 | 20 | 3 | 0.3 |

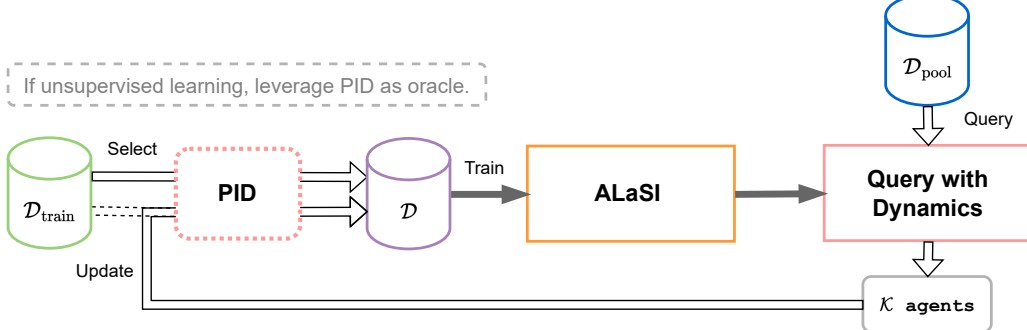

Figure 4: General pipeline of ALaSI.

## B.4    DETAILS OF LOSS FUNCTION

In this section, we discuss and state the details of loss terms and the implementation details of the proposed loss terms in hybrid loss 16.

### B.4.1    OOS LOSS

In this section, we describe the implementation of OOS loss function (Equation 12). As shown in Section A.2, the loss function is simplified as the maximization of mutual information between $Z_{\text{oos}}^t$ and $v_i^{t+1}$ for all $0 \leq t \leq T - 1$, and for all agent $i$ in the current scope. As mentioned in Section 4.3, we leverage DeepInfoMax (Hjelm et al., 2019) to maximize $I(Z_{\text{oos}}^t, v_i^{t+1})$. We follow the implementation of DeepInfoMax at: `https://github.com/DuaneNielsen/DeepInfomaxPytorch`, which is a pytorch version of official implementation at `https://github.com/rdevon/DIM`. Interestingly, DeepInfoMax requires output variables, input variables and also the negative samples of input variables. As a result, besides $Z_{\text{oos}}^t$ and $v_i^{t+1}$, we also feed $\mathcal{V}_{\text{inter}}^t$ to DeepInfoMax, as the negative samples.

### B.4.2    DISTANCE CORRELATION

In this section, we firstly describe the procedures to calculate distance correlation $\mathcal{L}_{\text{dc}}$ in Equation 13, then we describe the implementation of distance correlation in our work.

**Procedures.**    We firstly pair the $K$ samples of $Z_{\text{oos}}^t$ and $\mathcal{V}_{\text{inter}}^t$ as pairs: $(z_p, x_p)_{p \in K}$. Then we calculate the distance matrices $\mathbf{A}, \mathbf{B} \in \mathbb{R}^{K \times K}$ as:

$$\mathbf{A}_{pq} = ||z_p - z_q||_F, \text{ and } \mathbf{B}_{pq} = ||x_p - x_q||_F, \ p, q = 1, ..., K.$$

After that, we double center the distance matrices to get $\tilde{\mathbf{A}}_{pq}, \tilde{\mathbf{B}}_{pq}$:

$$\tilde{\mathbf{A}}_{pq} = \mathbf{A}_{pq} - \bar{\mathbf{A}}_{p.} - \bar{\mathbf{A}}_{.q} + \bar{\mathbf{A}}_{..},$$

where $\bar{\mathbf{A}}_{i.}$ denoted the mean of row $i$, $\bar{\mathbf{A}}_{.j}$ denotes the mean of column $j$, $\bar{\mathbf{A}}_{..}$ denotes the overall mean of $\mathbf{A}$. So this centers both the rows and columns of $A, B$. All rows and columns of $\tilde{\mathbf{A}}$ and $\tilde{\mathbf{B}}$

---

**Algorithm 1** Pipeline of ALaSI.

---

1: **Input:** $\mathcal{D}_{\text{train}}$ = a pool of labeled trajectories $\{\mathcal{V}_{\text{train}}, \mathcal{E}\}$,
2: **Input:** $\mathcal{D}_{\text{pool}}$ = a pool of test trajectories $\{\mathcal{V}_{\text{pool}}\}$,
3: **Parameters:** initial sample size $m$, query size $\mathcal{K}$, number of epochs $E$, number of selection rounds $N$,
4: **Model Weights:** $\theta$, **Hybrid Loss:** $\mathcal{R}$, **Query with Dynamics:** $\mathcal{Q}$,
5: **Output:** Trained Active Structural Inference Model $\mathcal{M}$,
6: **if** Supervised learning **then**
7:     Set of data points $\mathcal{D} \leftarrow$ Select $m$ agents with features $\mathcal{V}_m$ and connectivity $\mathcal{V}_m$ from $\mathcal{D}_{\text{train}}$,
8: **else**
9:     Select $m$ agents with features $\mathcal{V}_m$ from $\mathcal{D}_{\text{train}}$,
10:     Run PID on $\mathcal{V}_m$ and obtain connections between $m$ nodes: $\mathcal{E}_{PID0}$,
11:     Set of data points $\mathcal{D} \leftarrow \{\mathcal{V}_m, \mathcal{E}_{PID0}\}$,
12: **end if**
13: Train model $\mathcal{M}$ $E$ epochs with loss $\mathcal{R}$ on $\mathcal{D}_{\text{train}}$ and obtain parameters $\theta_0$,
14: Query $\mathcal{K}$ agents with the strategy of query with dynamics $\mathcal{Q}(\theta_0, \{\mathcal{V}_{\text{pool}}\}, \mathcal{E})$,
15: **if** Supervised learning **then**
16:     Update $\mathcal{D}$ with $\mathcal{K}$ agents with features $\mathcal{V}_{\mathcal{K}}$ and connectivity $\mathcal{V}_{\mathcal{K}}$ from $\mathcal{D}_{\text{train}}$,
17: **else**
18:     Select $m$ agents with features $\mathcal{V}_m$ from $\mathcal{D}_{\text{train}}$,
19:     Run PID on features $\mathcal{V}_{\mathcal{K}}$ and obtain connections between $\mathcal{K}$ nodes: $\mathcal{E}_{PIDK}$,
20:     Update $\mathcal{D} \leftarrow \{\mathcal{V}_{\mathcal{K}}, \mathcal{E}_{PIDK}\}$,
21: **end if**
22: **while** Round $i < N$ **do**
23:     Train model $\mathcal{M}$ $E$ epochs with loss $\mathcal{R}$ on $\mathcal{D}_{\text{train}}$ and obtain parameters $\theta_i$,
24:     Query agent features with $\mathcal{Q}(\theta_i, \{\mathcal{V}_{\text{pool}}\}, \mathcal{E})$ and choose $\mathcal{K}$ agents,
25:     **if** Supervised learning **then**
26:         Update $\mathcal{D}$ with $\mathcal{K}$ agents with features $\mathcal{V}_{\mathcal{K}}$ and connectivity $\mathcal{V}_{\mathcal{K}}$ from $\mathcal{D}_{\text{train}}$,
27:     **else**
28:         Select $m$ agents with features $\mathcal{V}_m$ from $\mathcal{D}_{\text{train}}$,
29:         Run PID on features $\mathcal{V}_{\mathcal{K}}$ and obtain connections between $\mathcal{K}$ nodes: $\mathcal{E}_{PIDK}$,
30:         Update $\mathcal{D} \leftarrow \{\mathcal{V}_{\mathcal{K}}, \mathcal{E}_{PIDK}\}$,
31:     **end if**
32: **end while**
33: Return trained model $\mathcal{M}$ and parameters $\theta$.

---

**Algorithm 2** Query with Dynamics $\mathcal{Q}$.

---

1: **Input:** $\mathcal{D}_{\text{train}}$ = a pool of labeled trajectories $\{\mathcal{V}_{\text{train}}, \mathcal{E}\}$,
2: **Input:** $\mathcal{D}_{\text{pool}}$ = a pool of test trajectories $\{\mathcal{V}_{\text{pool}}\}$,
3: **Input:** $\mathcal{D}$ = a pool of agents we have for training,
4: **Parameters:** Query Size: $\mathcal{K}$,
5: **Model Weights:** $\theta$, **Dynamic Loss:** $\mathcal{L}_{\text{D}}$,
6: **Output:** Query of $\mathcal{K}$ agents,
7: Calculate dynamics loss $\mathcal{L}_{\text{D}}$ on all of the agents in $\mathcal{D}_{\text{pool}}$ with only one other agent in scope,
8: Select $\mathcal{K}$ agents with largest dynamics prediction error,
9: Return $\mathcal{K}$ agents and update $\mathcal{D} = \mathcal{D} \cup \mathcal{V}_i, i \in \{\mathcal{K}\}$ with features and connectivity from $\mathcal{D}_{\text{train}}$.

---

sum to 0. In short notation:

$$\tilde{\mathbf{A}}_{qm} = (I - M)\mathbf{A}(I - M), \text{ and } \tilde{\mathbf{B}}_{qm} = (I - M)\mathbf{A}(I - M),$$

where $M = \frac{1}{K}\mathbf{1}\mathbf{1}^T$.

The distance covariance of $Z_{\text{oos}}^t$ and $\mathcal{V}_{\text{inter}}^t$ is defined as the square root of:

$$\text{dcov}^2(Z_{\text{oos}}^t, \mathcal{V}_{\text{inter}}^t) = \frac{1}{K^2} \sum_{p,q=1}^{K} \tilde{\mathbf{A}}_{pq}\tilde{\mathbf{B}}_{pq}.$$

---

**Algorithm 3** PID Algorithm in ALaSI.

---

1: **Input:** $\{\mathcal{V}_{\text{pool}}\}$ = a pool of trajectories of $p$ agents,
2: **Parameters:** Rank or proportion of rank: $\xi$, Total number of time steps of features: $T$,
3: **Output:** $\mathcal{D}_{\text{train}}$ = a pool of labeled trajectories $\{\mathcal{V}_{\text{pool}}, \mathcal{E}\}$,
4: **for** agent $i$ in total $p$ agents **do**
5:     **for** agent $j$ in $p-1$ agents **do**
6:         **for** agent $r$ in $p-2$ agents **do**
7:             Compute the unique component $I_{\text{Uni}}$ between $X_i^{1:T-1}$ and $X_j^{2:T}$ given $X_r^{2:T}$,
8:             Compute the mutual information $I$ between $X_i^{1:T-1}$ and $X_j^{2:T}$ given $X_r^{2:T}$,
9:             Compute the ratio $q_r$ between the $I_{\text{Uni}}$ and $I$,
10:         **end for**
11:         Calculate the sum of $q_r$ over all agents $r$ as $q_{ij}$,
12:     **end for**
13: **end for**
14: Rank all $q_{ij}$, and select $\xi$ (or $\xi \cdot p$) agent-pairs with highest $q_{ij}$,
15: Mark the connections from $i$ to $j$ in these pairs as exist, the rest as non-exist,
16: Return the connectivity between $p$ agents.

---

And the distance variance is defined as: $\text{dvar}^2(x) = \text{dcov}^2(x, x)$. Thus we can calculate the distance correlation with:

$$\mathcal{L}_{\text{dc}} = \frac{1}{(T-1) \cdot |\mathcal{D}|} \sum_t^{T-1} \sum_{i \in \mathcal{D}} \frac{\text{dCov}^2(Z_{\text{oos}}^t, \mathcal{V}_{\text{inter}}^t)}{\sqrt{\text{dVar}(Z_{\text{oos}}^t)\text{dVar}(\mathcal{V}_{\text{inter}}^t)}}.$$

**Implementation.** As for implementation of distance correlation, we originally follow the the official implementation of distance correlation implementation of Zhen et al. (2022) at `https://github.com/zhenxingjian/Partial_Distance_Correlation`. We then extend the implementation to suit the batch-wise calculation and the GPU acceleration.

## B.5 Implementation of Pipelines

We firstly briefly describe the pipeline of learning of ALaSI in Algorithm 4.

---

**Algorithm 4** Pipeline of learning in ALaSI.

---

1: **Input:** $\mathcal{V}$ = set of agent features of current scope,
2: **Input:** $n$ = number of agents in the current scope,
3: **Input:** $Z_{\text{gt}}$ = ground truth connectivity in the current scope,
4: Connection Learning Network: $f_{\text{inter1}}$, Inter-scope Message Network: $f_{\text{inter2}}$, OOS Embedding Network: $f_{\text{oos1}}$, OOS Message Network: $f_{\text{oos2}}$, Dynamics Learning Network: $f_{\text{dynamics}}$, Output Function: $f_{\text{output}}$, DeepInfoMax: $f_{\text{DIM}}$
5: Split agent features according to time steps: $\mathcal{V}_\tau = \mathcal{V}^{0:T-1}$ for training, $\mathcal{V}_\psi = \mathcal{V}^{1:T}$ for loss calculation, where $T$ represents the total time steps,
6: Learn representation of connections: $Z = f_{\text{inter1}}(\mathcal{V}_\tau, n)$,
7: Summarize connectivity inside the scope: $\hat{Z} = GumbelSoftmax(Z)$,
8: Learn inter scope messages: $e_{\text{inter}} = f_{\text{inter2}}(\mathcal{V}_\tau, \hat{Z})$,
9: Learn OOS messages: $e_{\text{oos}} = f_{\text{oos2}}(f_{\text{oos1}}(\mathcal{V}_\tau, e_{\text{inter}}))$,
10: Learn dynamics: $e_{\text{out}} = f_{\text{output}}(f_{\text{dynamics}}(e_{\text{inter}}, e_{\text{oos}}), \mathcal{V}_\tau)$,
11: Calculate OOS loss with DeepInfoMax: $\mathcal{L}_{\text{OOS}} = f_{\text{DIM}}(e_{\text{inter}}, e_{\text{oos}}, \mathcal{V}_\tau)$,
12: Calculate distance correlations: $\mathcal{L}_{\text{dc}}$ from $e_{\text{oos}}$ and $\mathcal{V}_\tau$ for each agent in the scope,
13: Calculate dynamics prediction loss: $\mathcal{L}_{\text{D}} \leftarrow \{e_{\text{out}}, \mathcal{V}_\psi\}$,
14: Calculate connectivity loss: $\mathcal{L}_{\text{con}} \leftarrow \{\hat{Z}, Z_{\text{gt}}\}$,
15: Summarize as the hybrid loss: $\mathcal{R} \leftarrow \{\mathcal{L}_{\text{D}}, \mathcal{L}_{\text{con}}, \mathcal{L}_{\text{OOS}}, \mathcal{L}_{\text{dc}}\}$,
16: Update parameters with back-propagation,
17: Return trained model.

---

We then describe the components of several networks mentioned in Algorithm 4. The design of $f_{\text{inter1}}$, $f_{\text{inter2}}$, $f_{\text{oos1}}$, $f_{\text{oos2}}$ and $f_{\text{dynamics}}$ follows modular-design practice and are based on a multi-layer-perceptron, and the multi-layer-perceptron is shown in Algorithm 5. The exact settings of the dimension of these layers may refer to our code in the supplementary document.

---

**Algorithm 5** The Multi-layer-perceptron.

---

1: **Input:** features $input$
2: x = elu(Linear1($input$))
3: x = dropout(x)
4: x = elu(Linear2(x))
5: out = batch_norm(x)
6: **Return:** out

---

We name the functional pipeline shown in Algorithm 5 as $MLP$, and we can represent the networks in Algorithm 4 as: $f_{\text{inter1}} = MLP(MLP(\cdot))$, $f_{\text{inter2}} = MLP(\cdot)$, $f_{\text{oos1}} = MLP(\cdot)$, $f_{\text{oos2}} = MLP(\cdot)$ and $f_{\text{dynamics}} = MLP(\cdot)$. We briefly report the dimension of the layers of each networks in Table 4, where $f''_{\text{inter1}}$ represents the second $MLP(\cdot)$ of $f_{\text{inter1}}$, $f'_{\text{inter1}}$ represents the first $MLP(\cdot)$ of $f_{\text{inter1}}$, $x_{dim}$ is the number of dimensions of an agent at a time step, and $|T|$ represents the total time steps of the trajectory.

Table 4: Dimension of the layers and dropout rates.

| Parameters | $f''_{\text{inter1}}$ | $f'_{\text{inter1}}$ | $f_{\text{inter2}}$ | $f_{\text{oos1}}$ | $f_{\text{oos2}}$ | $f_{\text{dynamics}}$ |
|---|---|---|---|---|---|---|
| Linear1 | $2 \cdot x_{dim} \cdot |T|$ | 256 | $2 \cdot x_{dim}$ | $(x_{dim} + 256) * (|T| - 1)$ | $2 \cdot x_{dim}$ | 256 |
| Dropout | 0.0 | 0.0 | 0.0 | 0.5 | 0.0 | 0.0 |
| Linear2 | 256 | 2 | 256 | $x_{dim} * (|T| - 1)$ | 256 | 256 |

## B.6 IMPLEMENTATION OF BASELINES

**NRI.** We use the official implementation code by the author from `https://github.com/ethanfetaya/NRI` with customized data loader for our chosen datasets. We add our metric-evaluation in "test" function, after the calculation of accuracy in the original code.

**fNRI.** We use the official implementation code by the author from `https://github.com/ekwebb/fNRI` with customized data loader for our chosen datasets. We add our metric-evaluation in "test" function, after the calculation of accuracy and the selection of correct order for the representations in latent spaces in the original code.

**MPM.** We use the official implementation code by the author from `https://github.com/hilbert9221/NRI-MPM` with customized data loader for our chosen datasets. We add our metric-evaluation for AUROC in "evaluate()" function of class "XNRIDECIns" in the original code.

**ACD.** We follow the official implementation code by the author as the framework for ACD (`https://github.com/loeweX/AmortizedCausalDiscovery`). We run the code with customized data loader for the chosen three datasets. We implement the metric-calculation pipeline in the "forward_pass_and_eval()" function.

**MPIR.** We follow the official implementation from `https://github.com/tailintalent/causal` as the model for MPIR. We run the model with customized data loader for the chosen datasets. After the obtain of the results, we run another script to calculate the metrics.

**PID.** Based on the Julia implementation of PID in `https://github.com/Tchanders/InformationMeasures.jl`, we implement PID in Python. Then we implement the mutual information calculation of PID with KDTree (see `https://github.com/paulbrodersen/entropy_estimators`), in order to enable PID to operate on continuous high-dimensional data. Different from other methods, we run PID on all of the dataset we have in experiments. For instance, when running experiments on "Springs50", PID infer the connections of the entire dynamical system based on a union set of the trajectories for training, validation and testing.

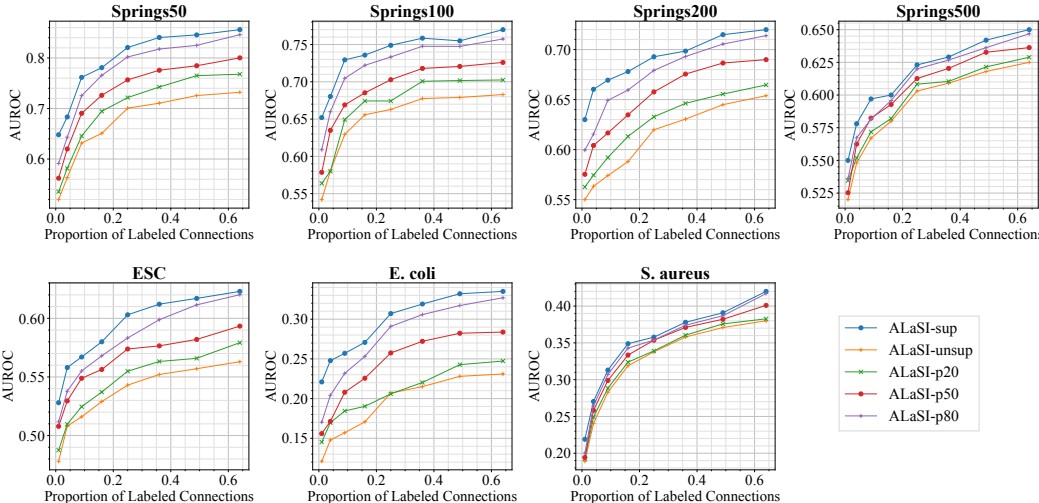

Figure 5: Averaged AUROC results of ALaSI-sup, ALaSI-unsup, ALaSI-p20, ALaSI-p50, and ALaSI-p80 as a function of the proportion of labeled connections on "Springs50", "Springs100", "Springs200", "Springs500", ESC, E. coli and S. aureus datasets.

## B.7 FURTHER DETAILS ABOUT DATASETS

**Spring Datasets** To generate these springs datasets ("Springs50", "Springs100", "Springs200", and "Springs500"), we follow the description of the data in Kipf et al. (2018) but with fixed connections. To be specific, at the beginning of the data generation for each springs dataset, we randomly generate a ground truth graph and then simulate 12000 trajectories on the same ground truth graph, but with different initial conditions. The rest settings are the same as that mentioned in Kipf et al. (2018). We collect the trajectories and randomly group them into three sets for training, validation and testing with the ratio of 8: 2: 2, respectively.

**GRN Datasets** Different from springs datasets, GRN datasets (ESC, E. coli, and S. aureus) are sampled from publicly available data sources. We download the datasets from the links mentioned in the corresponding literature, sample the trajectories with the same amount of time steps as of springs datasets, and randomly group the trajectories of gene expressions into three sets for training, validation and testing with the ratio of 8: 2: 2, respectively.

## C FURTHER EXPERIMENTAL RESULTS

In this section, we demonstrate additional experimental results as the supplement to Section 5.

### C.1 INTEGRATION OF PRIOR KNOWLEDGE WITH UNSUPERVISED LEARNING

We conduct the integration of prior knowledge with unsupervised learning with ALaSI. At the beginning of every experiment, we randomly assign a portion of agents with true connectivity, and keep the remaining settings the same in Section 5.2. During a query, if the agents with true connectivity are selected and the connections of these agents assigned by PID are contrary to the true label, we set the connectivity the same as the label and maintain the connections of the rest agents. We summarize the results and plot them in Figure 5, where we plot the AUROC results of fully supervised ALaSI (ALaSI-sup), fully unsupervised ALaSI (ALaSI-unsup), and unsupervised ALaSI with 20%, 50% and 80% of prior knowledge on agents (ALaSI-p20, ALaSI-p50 and ALaSI-p80). As we can observe from the plots, ALaSI is capable of being integrated with prior knowledge, and the AUROC value is positively correlated with the proportion of integrated prior knowledge. Interestingly, ALaSI-p80 moves generally closer to the fully supervised ALaSI, which on the other hand verifies the data efficiency of ALaSI. ALaSI has the capability of inferring accurate connectivity of dynamical systems with less prior knowledge. In comparison, we also tested the integration of prior

knowledge with baseline methods that uses VAE under unsupervised settings, but surprisingly we observed performance drops in terms of AUROC. We think the reason might be the integration of prior knowledge happened in the latent space, violating the generation process of these methods. We leave the study of these performance drops to future work.

## C.2 Robustness Tests of ALaSI

Although ALaSI is tested on several real-world datasets and the results are reported in Sections 5.1 and 5.2, it is interesting to carry out more experiments to further test the robustness of ALaSI. We generate a series of "Springs50" datasets with different level of Gaussian noise. The Gaussian noise is added to the features of the agents and the levels $\Delta$ amplify the noise as follows:

$$\tilde{v}_i^t = v_i^t + \zeta \cdot 0.02 \cdot \Delta, \text{ where } \zeta \sim \mathcal{N}(0, 1), \tag{23}$$

where $v_i^t$ represents raw feature vector of agent $i$ at time $t$. And we plot the experimental results of ALaSI on these datasets in Figure 6. As shown in Figure 6, noises in the agents' features have an

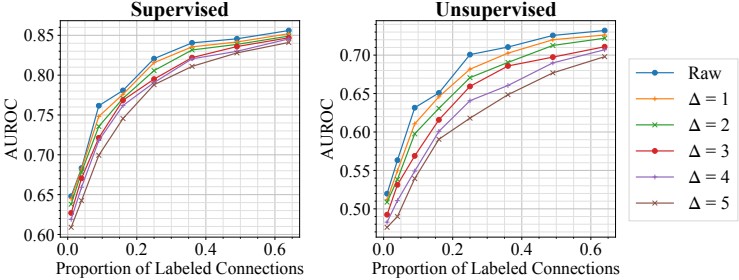

Figure 6: Averaged AUROC results of ALaSI as a function of the proportion of labeled connections on "Springs50" dataset with different levels of noise under supervised or unsupervised setting.

effect on the performance of ALaSI. The effect is minor when ALaSI is trained under supervised setting. But under unsupervised setting, especially when the proportion of labeled connections in the pool is smaller than 0.4, ALaSI faces bigger challenge to infer the connections compared with under supervised setting. When the proportion of labeled connections increases, the effect of noises become smaller and smaller. So in summary, although noises have negative impact on the performance, ALaSI still can infer the connections with moderate to high accuracy. Besides that, we also test the baseline methods on the dataset of "Springs50" with different levels of Gaussian noise, and plot the results in Figure 7. Each subplot in Figure 7 reports the performance of ALaSI and baseline methods on the "Springs50" dataset with a certain noise level, respectively. As we can learn from the figure, although the baseline methods are trained under supervised settings, compared to ALaSI, they are more sensitive to the noises. The margin between the AUROC results of ALaSI and the best baseline methods becomes larger when the noise level increases. We think the reason may come from that the baseline methods utilize a full-sized computational graph, so during training, all of the connections within the system are learned simultaneously. Therefore, high level of noise leads to an enormous uncertainty in the loss functions of these methods (their loss functions are summations of errors of all the connections in the system). Different from baseline methods, ALaSI learns the connections agent-wise, which ease the uncertainty in the loss function. Besides that, the query with dynamics can correctly select the most informative agent to be added to the scope, regardless of the noise level. We think a combination of these two functioning mechanism helps ALaSI to reduce the uncertainty created by noisy data.

## C.3 Limitation of ALaSI

Besides the datasets mentioned in this work, we also test ALaSI on the physic simulation datasets mentioned in NRI (Kipf et al., 2018). Most of the physic simulation datasets have no more than 10 agents in the system, which are much smaller than the ones used in this work. Based the experiments on these datasets, ALaSI cannot outperform baseline methods when the size of the dynamical system is small. Since ALaSI works on agent-wise selection to build the pool for training, when the total count of agents is small, ALaSI cannot be beneficial from the mechanism of active learning.

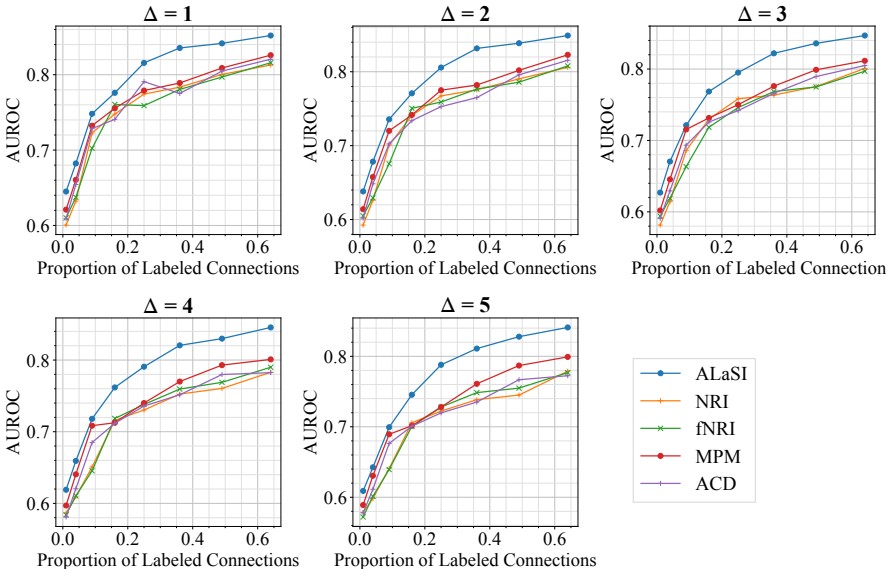

Figure 7: Averaged AUROC results of ALaSI and baseline methods as a function of the proportion of labeled connections on "Springs50" dataset with different levels of noise under supervised setting.

## D    ETHICS STATEMENT

ALaSI is a framework for structural inference of dynamical systems. No matter how effective it is at this task, there may still be failure-modes ALaSI will not catch. So far in this work we haven't seen any issue with ethics.

## E    REPRODUCIBILITY

We attach a link to our anonymous repository in the supplementary document. We include the codes of ALaSI, and the procedures for accessing the dataset we used in this work. Please refer to it as the implementation of ALaSI.

