# OpenReview forum: "Active Learning based Structural Inference"
_ICLR.cc/2023/Conference — Submitted to ICLR 2023_

### Official Review · Reviewer_o2om · 2022-10-25

**Confidence:** 3
**Correctness:** 3
**Technical Novelty And Significance:** 2
**Empirical Novelty And Significance:** 3
**Recommendation:** 6

**Clarity, Quality, Novelty And Reproducibility:**

This paper is well-written and generally easy to follow. From what I understand, the main novelty lies in the proposed inter- and out-of-scope operations, and other tricks (e.g., the proposed hybrid loss function) are standard or intuitive to the literature.

**Strength And Weaknesses:**

Strength:
1. While I'm not an expert on structural inference in dynamic systems, the proposed inter- and out-of-scope operations look interesting and novel. These operations help make the proposed framework scalable to large dynamic systems.
2. The authors run experiments on multiple datasets and compare ALsSI with multiple baselines. The proposed method outperforms all existing baselines on all datasets.

Weakness: Besides the proposed inter- and out-of-scope operations, other tricks, e.g., the hybrid loss and PID, studied in the paper seem to be standard to the literature or at least very intuitive.

**Summary Of The Paper:**

This paper studies how to leverage deep active learning to infer the existence of directed connections in dynamic systems. The authors propose a framework called Active Learning based Structural Inference (ALaSI). The authors design inter- and out-of-scope operations to make ALaSI scalable, and evaluate the framework on many datasets. Experimental results show the efficacy of the proposed framework.

**Summary Of The Review:**

While I am not quite familiar with the literature on structural inference, the experimental results show that the authors propose an effective method that outperforms all existing baselines. Based on that, I am inclined to accept the paper (but I'll set my confidence score at 3).

====after rebuttal====

I thank the authors for their response. I have read the response and would like to keep my scores. I also suggest the authors add more explanations to the paper to make it more readable (e.g., backgrounds/explanations on PID).

---

> ### Author Response · Authors · 2022-11-08
> **Response to Reviewer o2om**
>
> We want to thank the reviewer for the review. We are happy that the reviewer found the inter-scope and out-of-scope operations interesting and novel.
>
> Here is our answer to the concern raised by the reviewer:
>
> > Besides the proposed inter- and out-of-scope operations, other tricks, e.g., the hybrid loss and PID, studied in the paper seem to be standard to the literature or at least very intuitive.
>
> Many thanks for the comment. We partially agree with the comment. The hybrid loss is widely adopted by NRI [2], fNRI [3], ACD [4] and iSIDG [1]. So we agree that hybrid loss is standard to the literature. But PID [5] is the method that can infer the existence of the connections precisely, such as discussed in [6]. Yet the original version of PID can only infer undirected connections. We extended PID to infer directed connections with the time-series features, which was novel in this field. We included the pipeline of calculating PID in Algorithm 3 in the appendix.
>
> Actually, we also thought about how to search for other possible oracles for ALaSI, and what features these oracles should contain, such as scalability or low bias. We would like to leave this question for future work.
>
> Besides that, we would like to ask the reviewer for possible further suggestions to make our paper more readable and more inspiring, such as from the aspect of a researcher not in the field of structural inference. We realize that since the research topic of structural inference is relatively new, any suggestions or advice would be of great help. Many thanks in advance.
>
> **References**
>
> [1] Aoran Wang and Jun Pang. Iterative structural inference of directed graphs. In Advances in Neural Information Processing Systems (NeurIPS), volume 35, 2022.
>
> [2] Thomas Kipf, Ethan Fetaya, Kuan-Chieh Wang, Max Welling, and Richard Zemel. Neural relational inference for interacting systems. In Proceedings of the 35th International Conference on Machine Learning (ICML), pp. 2688–2697. PMLR, 2018.
>
> [3] Ezra Webb, Ben Day, Helena Andres-Terre, and Pietro Lio. Factorised neural relational inference ´ for multi-interaction systems. arXiv preprints arXiv:1905.08721, 2019.
>
> [4] Sindy Lowe, David Madras, Richard Z. Shilling, and Max Welling. Amortized causal discovery: ¨ Learning to infer causal graphs from time-series data. In Proceedings of the 1st Conference on Causal Learning and Reasoning (CLeaR), pp. 509–525. PMLR, 2022.
>
> [5] Paul L Williams and Randall D Beer. Nonnegative decomposition of multivariate information. arXiv preprint arXiv:1004.2515, 2010.
>
> [6] Aditya Pratapa, Amogh P Jalihal, Jeffrey N Law, Aditya Bharadwaj, and TM Murali. Benchmarking algorithms for gene regulatory network inference from single-cell transcriptomic data. Nature Methods, 17(2):147–154, 2020.

---

> ### Author Response · Authors · 2022-12-09
> **Response to the Update**
>
> Dear Reviewer o2om,
>
> Many thanks for the suggestions on making the paper clearer and more readable. We would like to reclaim that the problem background is the same as those in NRI [1], fNRI [2], and iSIDG [3]: to study the inference of connections between agent-pairs in a dynamical system. However, previous methods lack scalability, and we address this problem with the help of active learning, which indeed turns the curse of scalability into a benefit. Following this line of thought, we found the necessity of learning OOS representations, and the importance of query with dynamics strategy to make ALaSI more efficient.
>
> Concerning the part of PID, we admit that we only included a few descriptions in the very beginning, because the algorithm itself could not be counted as the contribution of this work, as it has been widely used in the field of biology in the inference of gene regulatory networks [4,5,6]. Despite our description in the appendix on the PID used in this work, we would also like to append more details about it in the camera-ready version. (Because now we cannot submit revisions at the moment.)
>
> We would like to thank Reviewer o2om again for the suggestions on our paper. It would be also great if we could have a discussion in the very first week of the rebuttal.
>
> Kind regards,
>
> Authors of Paper4510
>
>
> ####  References
>
> [1] Thomas Kipf, Ethan Fetaya, Kuan-Chieh Wang, Max Welling, and Richard Zemel. Neural relational inference for interacting systems. In Proceedings of the 35th International Conference on Machine Learning (ICML), pp. 2688–2697. PMLR, 2018.
>
> [2] Ezra Webb, Ben Day, Helena Andres-Terre, and Pietro Lio. Factorised neural relational inference for multi-interaction systems. arXiv preprints arXiv:1905.08721, 2019.
>
> [3] Aoran Wang and Jun Pang. Iterative structural inference of directed graphs. In Advances in Neural Information Processing Systems (NeurIPS), volume 35, 2022.
>
> [4] Michael Wibral, Viola Priesemann, Jim W. Kay, Joseph T. Lizier, and William A. Phillips. Partial information decomposition as a unified approach to the specification of neural goal functions. Brain and Cognition, Volume 112, 2017, Pages 25-38.
>
> [5] Joseph T. Lizier, Benjamin Flecker, and Paul L. Williams. Towards a synergy-based approach to measuring information modification. In 2013 IEEE Symposium on Artificial Life (ALIFE), pp. 43-51. IEEE, 2013.
>
> [6] Ernesto Pereda, Quiroga Quian Rodrigo, and Joydeep Bhattacharya. Nonlinear multivariate analysis of neurophysiological signals. Progress in neurobiology 77, no. 1-2 (2005): 1-37.

---

### Official Review · Reviewer_5BCf · 2022-10-25

**Confidence:** 3
**Correctness:** 4
**Technical Novelty And Significance:** 3
**Empirical Novelty And Significance:** 3
**Recommendation:** 6

**Clarity, Quality, Novelty And Reproducibility:**

- Clarity: The paper is well written.
- Quality: Empirical and theoretically solid.
- Novelty: The proposed method of active learning for agent-wise selection with inter-scope and out-of-scope operations are novel.
- Reproducibility: The authors have provided implementation details in Section B in the appendix.

**Strength And Weaknesses:**

#### Strengths
- Overall the paper is well written and easy to understand.
- The inter-scope and out-of-scope operations are well-designed.
- The experimental validation is very solid and convincing.

#### Weakness
- The current manuscript lacks the discussion of limitations or failure cases.


**Summary Of The Paper:**

The paper proposes a scalable structure inference framework based on agent-wise DeepAL called ALaSI to actively select the most informative agents to be labeled based on dynamic errors. In particular, with the inter-scope and out-of-scope operations,  ALaSI is able to distinguish between inter-scope messages and OOS messages based on the current view of the partial system. The experiments conducted under supervised settings seem to validate the efficacy of the proposed method.

**Summary Of The Review:**

Based on the above statements, I would like to weakly accept the paper initially. After carefully reading the other reviewers' comments and the authors' response, I would like to change my rating from "8: accept, good paper" to "6: marginally above the acceptance threshold" due to the concerns of novelty and experimental validation raised by the other reviewers.

---

> ### Author Response · Authors · 2022-11-08
> **Response to Reviewer 5BCf**
>
> We want to thank the reviewer for the motivating review. We are happy the reviewer found our method ALaSI novel and the paper well-written.
>
> Here is our answer to the concern raised by the reviewer:
>
> > The current manuscript lacks the discussion of limitations or failure cases.
>
> Good question! Based on our previous experiments, ALaSI could not outperform baseline methods when the size of the dynamical system is small. For example, we tested ALaSI and the baseline methods on the datasets mentioned in [1], which only contain systems with no more than 10 agents. And ALaSI could not outperform baseline methods. Since ALaSI works on agent-wise selection to build the pool for training, when the total count of agents is small, ALaSI cannot be beneficial from the mechanism of active learning. We think this may be the limitation of ALaSI. We will include the limitation in the appendix of the revision. (We cannot fit it in the main body of the paper.)
>
>
> **Reference**
>
> [1] Aoran Wang and Jun Pang. Iterative structural inference of directed graphs. In Advances in Neural Information Processing Systems (NeurIPS), volume 35, 2022.

---

> ### Author Response · Authors · 2022-12-09
> **Response to the Update**
>
> Dear Reviewer 5BCf,
>
> We are sad to see that you changed your mind and changed the rating to 6. We would like to re-claim that we do not agree with the comments on the novelty and experimental validation raised by the other reviewers.
>
> As you may have read in our responses to Reviewer Sh3s,  we stated the novelty of this work to be solving the problem of scalability in structural inference with the help of active learning, which actually turns the curse of scalability into benefits. Following this line of thought, we studied and proposed OOS representation learning pipeline and the query with dynamics strategy to enable the active learning pipeline on structural inference, and both of which are novel. As a result, we were confused when Reviewer Sh3s claimed that this work was not novel in his review.
>
> Besides that, we believe that our validation of ALaSI on simulation datasets, and on real-world datasets in the initial submission, as well as the appended extra experiments on datasets with different levels of noise in the revision strongly support the performance of ALaSI. As we are still looking forward to the response of Reviewer Sh3s, we would also like to sincerely invite you to join the discussion, in order to address the shortcomings of this paper in more detail. We are open to all kinds of encouragement and suggestions on this paper.
>
> Many thanks again for your efforts in the reviewing process.
>
> Kind regards,
>
> Authors of Paper4510

---

### Official Review · Reviewer_Sh3s · 2022-10-27

**Confidence:** 3
**Correctness:** 3
**Technical Novelty And Significance:** 2
**Empirical Novelty And Significance:** 2
**Recommendation:** 3

**Clarity, Quality, Novelty And Reproducibility:**

Clarity
This work is not self-contained (see weakness 5) and is difficult to follow. For example, PID (partial information decomposition) is an important concept in section 3, but the authors start to explain it very late in section 4.5. What are the model parameters in equation 1? Why choose this particular form? How does this connect to equation 4?

Quality
This loss of the ALaSI is complicated and I am also concerned how this method works on trajectory prediction. I think this will make the method more convincing.

Novelty
Structural Inference is not new, but it is a novel problem in active learning.

Reproducibility
The algorithm is clear in the appendix, but the authors do not provide how they set the parameters and the hyper parameters, such as learning rate, hidden dimensions, network layers, etc.


**Strength And Weaknesses:**

Strength
1.	The authors explore a novel problem in active learning.
2.	The authors consider the problem comprehensively. They scale to large data size and the loss can help learn OOS representation, estimating dynamic error and connectivity prediction error.

Weakness
1.	This paper is not well written and not self-contained. A lot of jargons are used without definition or clarification, such as inter and out scope, pool, query with dynamics.

2.	The loss of ALaSI is complicated with four parts, and each part has its own tuning parameter. The paper does not provide a good strategy to tune the alpha, beta, gamma and eta, which will make it hard to search an optimal combination of the parameters. The authors should report ablation results to convince the reader that each part of the loss plays an important role.

3.	The authors only show the results of the link prediction, which is not sufficient for structural inference. The authors should plot a figure showing that the predicted trajectory is closed to the ground truth, like in other papers such as NRI, fNRI, ACD.

4.	While titled active learning based structural inference, this paper is light on the active learning part. Only section 4.4 briefly covered AL.  There is not sufficient details to understand how the AL works with the rest system.

5.	The authors do not show whether the ALaSI method is robust to noisy data. And the performance under different noise level.

6.	Minor: in figure 2, the second figure in the first row, why does the AUROC decrease between 0.3 and 0.5 proportion of labeled connection? With more proportion of labeled data, the AUROC should increase.


**Summary Of The Paper:**

In this paper, the authors explore a novel problem structural inference in active learning. They proposed ALaSI to infer the existing connections in the dynamic system in both supervised learning and unsupervised learning. The ALaSI can distinguish inter and out of scope information and its loss is built on three goals
i) motivating model to extract OOS representations,
ii) computing dynamic error,
and iii) estimating the connectivity prediction error.
The experiment results on seven dynamic systems show that ALaSI can infer the connection better than the state-of-the-art methods.
But the paper is not well written, lacking of consistency and clarification.


**Summary Of The Review:**

This paper needs to address the aforementioned concerns. Please refer to specific  comments above.

---

> ### Author Response · Authors · 2022-11-08
> **Response to Reviewer Sh3s (Part 1)**
>
> First of all, we want to thank the reviewer for the thoughtful and helpful review. We are happy the reviewer found ALaSI to be a novel approach in structural inference and can scale to large data size.
>
> Here are our answers to the concerns raised by the reviewer:
>
> > This paper is not well written and not self-contained. A lot of jargons are used without definition or clarification, such as inter and out scope, pool, query with dynamics.
>
> Many thanks for the comment. We tried to express our idea with some easy-to-understand nicknames, and meanwhile, we also tried to search for corresponding phrases in the field of active learning, which resulted in the "jargon" such as "inter- and out-of-scope", "pool" and "query with dynamics".
>
> Before we talk about the definition of inter- and out-of-scope in our paper, it is essential to know what is a scope. We gave the definition that partial knowledge on connectivity is called a scope, and this can be found in the last paragraph of Section 1 (the first line on page 2), and in the first paragraph of Section 4.2 (on page 4). In other words, a scope represents a partial view of the dynamical system, and consists of a set of features and connectivity of agents inside the partial system.
>
> Therefore, we can group the agents from the whole system into two clusters: the agents inside the partial system, and the agents outside of the partial system. And we can group the connections between the agents into three groups according to the position of the agents:
>
> - Inter-scope connections: the connections that both ends are agents inside the partial system; (the definition can be found in the first paragraph of Section 4.2)
> - Out-of-scope connections: the connections that connect one agent in the partial system, and the other agent outside of the partial system; (the definition can be found in the first paragraph of Section 4.2)
> - Unobservable connections: both agents are not in the partial system.
>
> As for the term "pool", it is used to represent a collection of data in the context of active learning, and the term is widely used in the literature on active learning and deep active learning. The list of literature consists of [1, 2, 3], and an exact definition of "pool" can be found in Section 1.2 of [4].
>
> Query strategies are important for active learning algorithms and the term is also widely used among the literature of active learning. Query strategies select the most informative samples from the pool, which will result in an efficient usage of available data. Then the active learning model will be trained with the samples. A raw definition of query strategy can be found in Section 1.2 of [4]. In our paper, the term "query with dynamics" refer to the strategy of ALaSI which is designed upon the dynamics prediction error. And the mechanism of "query with dynamics" can be found in Section 4.4 of our paper.

---

> > ### Author Response · Authors · 2022-11-08
> > **Response to Reviewer Sh3s (Part 2)**
> >
> > >The loss of ALaSI is complicated with four parts, and each part has its own tuning parameter. The paper does not provide a good strategy to tune the alpha, beta, gamma and eta, which will make it hard to search an optimal combination of the parameters. The authors should report ablation results to convince the reader that each part of the loss plays an important role.
> >
> > Many thanks for the comment. Yes, we agree that the method of searching for the optimal combination of the parameters in the hybrid loss is not mentioned in the paper. We reported the choice of parameters based on the values that can match all of the terms into the same scale. And even based on these easy searches, ALaSI managed to outperform other baseline methods. We think that it is feasible to tune these parameters with the help of Bayesian Optimization packages, such as [HEBO](https://github.com/huawei-noah/HEBO), or [TuRBO](https://github.com/uber-research/TuRBO).
> >
> > We may have a close look at the terms in the hybrid loss mentioned in Section 4.3. The hybrid loss consists of four terms,  \\(\mathcal{L}\_{oos}\\): to learn OOS messages, \\(\mathcal{L}\_{dc}\\): to ensure the independence assumption of learning OOS messages, \\(\mathcal{L}\_D\\): loss function for dynamics, and \\(\mathcal{L}\_{con}\\): loss function for connectivity. Among the four terms, \\(\mathcal{L}\_{oos}\\) and \\(\mathcal{L}\_{dc}\\) should appear in pairs to learn OOS messages (as we stated and proved in Section A.2 in the appendix). \\(\mathcal{L}\_D\\) and \\(\mathcal{L}\_{con}\\) are very important terms to make ALaSI work (terms for AL training), which cannot be discarded. Therefore, we conducted ablation studies to check the importance of terms for OOS message learning and presented the results in Section 5.3. In the ablation studies, we state "ALaSI-no OOS" as the one without \\(\mathcal{L}\_{oos}\\) and \\(\mathcal{L}\_{dc}\\) by setting \\(\gamma\\) and \\(\eta\\) as zero. And Figure 3 clearly shows the importance of \\(\mathcal{L}\_{oos}\\) and \\(\mathcal{L}\_{dc}\\). Without these two terms, the algorithm can only learn about the representations of connections within the scope and cannot extrapolate onto OOS connections, which results in an almost linear dependence between AUROC and the proportion of labeled connections.
> > > The authors only show the results of the link prediction, which is not sufficient for structural inference. The authors should plot a figure showing that the predicted trajectory is close to the ground truth, like in other papers such as NRI, fNRI, ACD.
> >
> > Many thanks for the suggestion. Actually, as stated in the second paragraph of Section 2, and the first paragraph of Section 3, we are interested in the reconstruction of the connections between the agents in a dynamical system with observational agents’ states. And that is the reason we call it "**Structural Inference**" instead of dynamics prediction. Besides that, not all of the baselines in the paper are able to output predict trajectories. For example, MPIR is a method based on minimum predictive information regularization, and does not output the future states of the agents. Indeed we will consider plotting a figure to show the predicted trajectories. But since the datasets in this work have far more agents (>50) than the ones in NRI [8], fNRI [9], ACD [10] (mostly less than 7), maybe we will plot the trajectories of only a few agents of the systems, otherwise, the plot would be confusing with too many trajectories.
> >
> > > While titled active learning based structural inference, this paper is light on the active learning part. Only section 4.4 briefly covered AL. There is not sufficient details to understand how the AL works with the rest system.
> >
> > We do not agree with this comment. We first formulate ALaSI in Equation 2 and Equation 3 in Section 3, which are basic formulations of active learning. We then describe how to formulate the learning problem of structural inference with the help of dynamics prediction in Equation 4. After that, we extend Equation 4 to cases with active learning, and show it in Equation 5. Since we have to build a pool of partial system to train ALaSI, the influence of inter-scope and OOS connections cannot be neglected. Following this line, we discuss how to learn inter-scope and OOS messages in Section 4.2. We then describe the hybrid loss in Section 4.3, which contains important terms for OOS message learning and also terms for basic AL training. After Section 4.4, we discuss the feasibility of unsupervised AL with the help of PID in Section 4.5. To conclude, we cover AL from Section 3 to Section 4.5.

---

> > > ### Author Response · Authors · 2022-11-08
> > > **Response to Reviewer Sh3s (Part 3)**
> > >
> > > > The authors do not show whether the ALaSI method is robust to noisy data. And the performance under different noise level.
> > >
> > > We do not agree with this comment. All of the GRNs (ESC, E.coli and S.aureus) in our paper are real-world datasets. The GRNs were sampled from experiments with unavoidable bias and noise.
> > >
> > > But we are still thankful for the advice of setting up datasets under different noise levels. If intended, we would like to generate a series of "Springs50" with different levels of Gaussian noise and plot the results in the appendix of revision.
> > >
> > > > Minor: in figure 2, the second figure in the first row, why does the AUROC decrease between 0.3 and 0.5 proportion of labeled connection? With more proportion of labeled data, the AUROC should increase.
> > >
> > > Good question! We ran the experiments ten times and reported the averaged results as the plots in Figure 2. The AUROC decrease at 0.5 proportion of labeled connection was caused by the performance drop of two runs. We checked the logs of these two runs, and interestingly found that the connectivity of the scopes was sparse, which meant that the category sets of connections and non-connections were strongly unbalanced. At this point, the strong imbalance negatively affected the training, and led to a performance drop. After this performance drop, the query with dynamics effectively updates the training pool by adding more informative connections to the scope, and manages to get rid of the imbalance.
> > >
> > > > For example, PID (partial information decomposition) is an important concept in section 3, but the authors start to explain it very late in section 4.5.
> > >
> > > We would like to argue that although our paper includes a modified PID to infer directed connections, the modification is just minor compared with the work that proposed PID [5]. Besides that, the modified PID is not our main contribution. And we think that it would be of great help if we first describe the functioning mechanism of ALaSI before PID. Otherwise, it could be more confusing. Therefore, we only briefly discuss PID in Section 4.5. For more details about the modified PID in our paper, please refer to Algorithm 3 in the appendix.
> > >
> > > > What are the model parameters in equation 1? Why choose this particular form? How does this connect to equation 4?
> > >
> > > Equation 1 is used to model the dynamical systems. It is not a machine learning model. In a broader view, dynamical systems are characterized by two main entities: the states of the nodes in the system, and the state transition functions of the nodes’ states. And these two can be unified and modeled as Equation 1, which is also mentioned in [6, 7].
> > >
> > > Equation 4 is a summarized version of machine learning mechanisms used in the work of NRI [8], fNRI [9] and ACD [10]. In this equation, \\(P(\hat{v}\_i^{t+1}|v^t\_i, \mathcal{U}\_i, \theta)\\) represents the machine learning model in those works, which tries to predict the future states based on observable present states. However, according to the formulation in Equation 1, an incorrect reconstruction of connectivity certainly leads to a large prediction error and is proved in [6]. So the models following the form Equation 4 will update their parameters to infer the connectivity of the system with an error that is as smaller as possible.
> > >
> > > > This loss of the ALaSI is complicated and I am also concerned how this method works on trajectory prediction. I think this will make the method more convincing.
> > >
> > > We already discussed the terms in the hybrid loss in the answers before. Since the focus of this work is the reconstruction of the connectivity of dynamical systems, we do not take trajectory prediction into account. As we show in Section 5, ALaSI outperforms all baseline methods on all of the datasets, even with real-world datasets. We think the results can be strong support for the performance of ALaSI on the task of structural inference. But we argue that it is feasible to use ALaSI for trajectory prediction by outputting the \\(\hat{v}^{t+1}\_i\\) term in Equation 5 as the prediction of every agent.
> > > > Structural Inference is not new, but it is a novel problem in active learning.
> > >
> > > Yes, we agree with this comment. Besides it is a novel problem in active learning, ALaSI is also a novel structural inference framework that can work on large dynamical systems.
> > >
> > > > The algorithm is clear in the appendix, but the authors do not provide how they set the parameters and the hyper parameters, such as learning rate, hidden dimensions, network layers, etc.
> > >
> > > Many thanks for the suggestion. The details of the implementation can be found in the anonymous repository we attached in the supplementary document. We are working on revision to include these parameters and we will include them in the appendix.

---

> > > > ### Author Response · Authors · 2022-11-08
> > > > **Response to Reviewer Sh3s (References)**
> > > >
> > > > **References**
> > > >
> > > > [1] Jordan T. Ash, Chicheng Zhang, Akshay Krishnamurthy, John Langford, and Alekh Agarwal. Deep batch active learning by diverse, uncertain gradient lower bounds. In Proceedings of the 8th International Conference on Learning Representations (ICLR), 2020.
> > > >
> > > > [2] Yarin Gal, Riashat Islam, and Zoubin Ghahramani. Deep bayesian active learning with image data. In Proceedings of the 34th International Conference on Machine Learning (ICML), pp. 1183– 1192. PMLR, 2017.
> > > >
> > > > [3] H. M. Sajjad Hossain and Nirmalya Roy. Active deep learning for activity recognition with context aware annotator selection. In Proceedings of the 25th ACM SIGKDD International Conference on Knowledge Discovery & Data Mining (KDD), pp. 1862–1870. ACM, 2019.
> > > >
> > > > [4] Burr Settles. Active Learning Literature Survey. Technical Report. University of Wisconsin-Madison Department of Computer Sciences, 2009.
> > > >
> > > > [5] Paul L Williams and Randall D Beer. Nonnegative decomposition of multivariate information. arXiv preprint arXiv:1004.2515, 2010.
> > > >
> > > > [6] Aoran Wang and Jun Pang. Iterative structural inference of directed graphs. In Advances in Neural Information Processing Systems (NeurIPS), volume 35, 2022.
> > > >
> > > > [7] Edward Beltrami. Mathematics for dynamic modeling. Academic press, 2014 May 10.
> > > >
> > > > [8] Thomas Kipf, Ethan Fetaya, Kuan-Chieh Wang, Max Welling, and Richard Zemel. Neural relational inference for interacting systems. In Proceedings of the 35th International Conference on Machine Learning (ICML), pp. 2688–2697. PMLR, 2018.
> > > >
> > > > [9] Ezra Webb, Ben Day, Helena Andres-Terre, and Pietro Lio. Factorised neural relational inference ´ for multi-interaction systems. arXiv preprints arXiv:1905.08721, 2019.
> > > >
> > > > [10] Sindy Lowe, David Madras, Richard Z. Shilling, and Max Welling. Amortized causal discovery: ¨ Learning to infer causal graphs from time-series data. In Proceedings of the 1st Conference on Causal Learning and Reasoning (CLeaR), pp. 509–525. PMLR, 2022.

---

> ### Author Response · Authors · 2022-11-14
> **Looking Forward to Discussion!**
>
> Dear Reviewer Sh3s,
>
> We hope our responses have answered your concerns about our work. We also revised our paper with additional experiments on the robustness test of ALaSI. Because of the page limitation, we only managed to merge the experiments in the appendix.
>
> If you have further questions about our paper, please let us know. We are also looking forward to an interactive discussion, which can deepen the comprehension of our work.
>
> Kind regards,
> Authors of Paper 4510

---

### Author Response · Authors · 2022-11-09
**Revision Summary**

We are thankful for the comments and suggestions from the reviewers, and we revised our paper with the following contents:

- Many thanks to the question of Reviewer Sh3s about the parameters of ALaSI. We added the description of the learning rate in Section B.1 of the appendix, and added the dimensions of layers in the networks in Section B.5 of the appendix.

- We would like to thank Reviewer Sh3s again for the advice of carrying out experiments on datasets with different noise levels, which will make ALaSI more convincing. Besides the experimental results on real-world GRNs in Section 5, we generated a new set of "Springs50" datasets with 5 different Gaussian noise levels, and we reported the results in Section C.2 of the appendix. As shown in the results, noise has a negative impact on the performance of ALaSI, but the impact is minor under supervised learning. When ALaSI is trained under unsupervised learning, the noises become more challenging, yet ALaSI still manages to infer the connections with moderate accuracy.

- Many thanks to the question of Reviewer 5BCf on the limitations of ALaSI. Yes, ALaSI definitely has limitations, and we added Section C.3 to describe the limitation of ALaSI.

- (New on Nov. 14, 2022) We summarized the results of baseline methods and ALaSI on "Springs50" with different levels of Gaussian noises. And we added the results and discussion in Section C.2 of the appendix.

We will update this comment when any revision is made.
We hope our responses to the comments and the revision can answer the questions of the reviewers.

---

### Author Response · Authors · 2022-11-18
**Follow-up**

Dear Reviewers,

How time flies, and there are only a few hours left in the discussion. We would really appreciate it if you could tell us if your concerns are resolved. We would be more than happy to resolve any remaining questions in the next few hours and would appreciate it if you engage in a discussion with us.

And we would try to revise the paper as soon as we can if you have any suggestions in the next few hours. After tomorrow if you have further suggestions or advice, we will try to revise the paper accordingly and upload it as a camera-ready version.

Kind regards,
Authors of Paper 4510

---

### Decision · Program_Chairs · 2023-01-20

**Decision:**

Reject

**Justification For Why Not Higher Score:**

This paper probably could be accepted — I don't think there is anything "wrong" — but reviewers had quite a bit of trouble with it, at least in part due to presentation. Additionally, most reviewers found the work to be somewhat "intuitive" (albeit in a less-studied area).

**Justification For Why Not Lower Score:**

N/A

**Metareview: Summary, Strengths And Weaknesses:**

This paper introduces a model for structural inference in dynamical systems, which also includes an active learning scheme. The paper introduces novel "inter-scope" and "out-of-scope" operations, and positive experimental results. Reviewers found some difficulty in following the paper; this is at least in part due to the paper's organization, which could be improved. This includes things like clearly defining terms that might be unfamiliar to machine learning reviewers. The contribution could also be made more clear: the paper introduces both a new model and an active learning scheme simultaneously, which makes it difficult to precisely judge either.

Originally, one of the reviewers argued to accept, one to reject, and one borderline. This paper was selected as a borderline paper which required additional discussion among the reviewers. The discussion is outlined in the next section. The authors added additional experiments after submission, which were helpful, but not sufficient to change the scores of the reviewers.



**Summary Of Ac-Reviewer Meeting:**

- On the positive side, this is a comparatively less-studied problem, and the method outperforms existing results
- There were not many technical contributions in the paper, in that to some degree the approach involved combining previously existing methods. This was cited by two reviewers (i.e. the degree of "novelty" present)
- The authors added additional experiments after submission, which were helpful, but not sufficient to change the scores of the reviewers.
- The authors discussed the many terms in the hybrid loss in more detail in the response, which is useful, and should go in the main paper

The consensus was that this paper had already been improved quite a bit from the original submission, but could handle some degree of reorganizing; this would probably require significant changes for acceptance that would necessitate further review after resubmission to another venue.

Ultimately, for a paper with borderline scores it is necessary for one of the reviewers to advocate for the paper and strongly argue acceptance; none of the reviewers did in this case.